# Coupled equilibria of dimerization and lipid binding modulate SARS Cov 2 Orf9b interactions and interferon response

CJ San Felipe[1], Jyoti Batra[2,3,4], Monita Muralidharan[2,3,4], Shivali Malpotra[2,3,4], Durga Anand[2,3,4], Rachel Bauer[2,4], Kliment A Verba[2,4], Danielle L Swaney[2,3,4], Nevan J Krogan[2,3,4], Michael Grabe[5]*, James S Fraser[1,2]*

[1]Department of Bioengineering and Therapeutic Sciences, University of California, San Francisco, San Francisco, United States; [2]Quantitative Biosciences Institute, University of California, San Francisco, San Francisco, San Francisco, United States; [3]Gladstone Institute of Data Science and Biotechnology, J. David Gladstone Institutes, San Francisco, United States; [4]Department of Cellular and Molecular Pharmacology, University of California, San Francisco, San Francisco, United States; [5]Department of Pharmaceutical Chemistry, University of California, San Francisco, San Francisco, United States

*For correspondence:
michael.grabe@ucsf.edu (MG);
jfraser@fraserlab.com (JSF)

## eLife Assessment

This **fundamental** study demonstrates that lipid binding can regulate the dimerization state of the SARS-CoV2 Orf9b protein. The data from biophysical and cellular experiments along with mathematical modeling are **compelling**. This paper is broadly relevant to those studying coupled equilibria across all aspects of biology.

**Abstract** Open Reading Frame 9b (Orf9b), an accessory protein of SARS-CoV and –2, is involved in innate immune suppression through its binding to the mitochondrial receptor Translocase of Outer Membrane 70 (Tom70). Previous structural studies of Orf9b in isolation revealed a β-sheet-rich homodimer; however, structures of Orf9b in complex with Tom70 revealed a monomeric helical fold. Here, we developed a biophysical model that quantifies how Orf9b switches between these conformations and binds to Tom70, a requirement for suppressing the type 1 interferon response. We used this model to characterize the effect of lipid binding and mutations in variants of concern to the Orf9b:Tom70 equilibrium. We found that the binding of a lipid to the Orf9b homodimer biases the Orf9b monomer:dimer equilibrium towards the dimer by reducing the dimer dissociation rate ~100 fold. We also found that mutations in variants of concern can alter different microscopic rate constants without significantly affecting binding to Tom70. Together, our results highlight how perturbations to different steps in these coupled equilibria can affect the apparent affinity of Orf9b to Tom70, with potential downstream implications for interferon signaling in coronavirus infection.

## Introduction

Activation of the innate immune system is a critical step to defending host cells from pathogens such as viruses (*Takeuchi and Akira, 2009*). One feature of the innate immune system is the activation of the interferon response (*Stetson and Medzhitov, 2006*). Upon viral infection, viral RNA is detected in the cytosol through intracellular MDA5/RIG-I-like receptors (*Pichlmair et al., 2006*; *Kang et al.,*

*2002*), which triggers the localization of the mitochondrial antiviral signaling (MAVS) protein to the mitochondrial outer membrane (*Seth et al., 2005*). At the membrane, MAVS functions as a scaffold for the recruitment of additional adaptor proteins that promote the activation of transcription factors such as IRF3 that localize to the nucleus and drive the expression of innate immune genes such as IFN-β (*Seth et al., 2005*) inducing an antiviral state. A prominent feature of both SARS-CoV and SARS-CoV-2 is that viral infection triggers a weak type 1 interferon response (IFN-I; *Blanco-Melo et al., 2020*). Several studies have illustrated the roles of both viral non-structural and accessory proteins in antagonizing different aspects of the innate immune response during viral infection (*Wang et al., 2021*; *Han et al., 2022*; *Felgenhauer et al., 2020*; *Shemesh et al., 2021*; *Liu et al., 2021*).

Open Reading Frame 9b (Orf9b) is an 11 kDa protein encoded through an alternative open reading frame within the N gene of both SARS-CoV and SARS-CoV-2. Orf9b localizes to the outer mitochondrial membrane during SARS-CoV and 2 infection (*Meier et al., 2006*; *Gordon et al., 2020*). Large-scale interactome profiling of SARS-CoV-2 proteins identified the mitochondrial import receptor Tom70 (Translocase of Outer Membrane 70) as interacting with Orf9b (*Gordon et al., 2020*). As part of the TOM complex, Tom70 facilitates translocation of cytoplasmic proteins across the mitochondrial outer membrane and the biogenesis of mitochondrial membrane proteins in a chaperone-dependent manner (*Backes et al., 2018*; *Backes et al., 2021*). In addition, Tom70 has also been identified as an important adaptor that binds MAVS and TBK1, leading to the induction of IFN-β (*Lin et al., 2010*; *Wei et al., 2015*; *Liu et al., 2010*). Both SARS-CoV and SARS-CoV-2 Orf9b have been shown to bind to Tom70 during infection, and this is sufficient to suppress the IFN response in the absence of additional viral factors (*Jiang et al., 2020*; *Gordon et al., 2020*). Therefore, Tom70 is an important target for Orf9b-mediated suppression of the innate immune response.

Tom70 possesses two distinct binding sites: a large C-terminal groove for the binding of amphipathic helices that form mitochondrial targeting sequences (MTS; *Young et al., 2003*; *Backes et al., 2018*) and an N-terminal chaperone binding site that recognizes the EEVD motif of Hsp70/90 (*Young et al., 2003*). The loss (either by truncation or mutation) of either of these binding sites impairs the import of proteins to the mitochondria (*Backes et al., 2021*). In the context of IFN signaling, the loss of the chaperone binding site is sufficient to suppress IFN signaling (*Lin et al., 2010*; *Wei et al., 2015*; *Liu et al., 2010*). Cryo-EM and X-ray structures of SARS-CoV-2 Orf9b:Tom70 showed Orf9b as a monomeric amphipathic helix bound to Tom70 at the C-terminal MTS binding site in a 1:1 interaction (*Gordon et al., 2020*; *Gao et al., 2021*). This was surprising because the crystal structures of both SARS-CoV (*Meier et al., 2006*) and SARS-CoV-2 (*Jin et al., 2023*, 7YE7, 7YE8, and unpublished structure 6Z4U) show Orf9b adopting a β-sheet-rich homodimer. This suggests that there is an equilibrium between the two Orf9b conformations (monomeric α-helix and β-sheet homodimer), but it is not clear how this equilibrium relates to IFN suppression or is regulated during viral infection.

X-ray and Cryo-EM structures of the Orf9b:Tom70 complex show that a key residue in the N-terminal chaperone binding site of Tom70 (R192) is displaced, which was hypothesized to impair binding of chaperones during IFN signaling. This allosteric conformational change is relevant because the kinase TBK1 binds to Tom70 in an Hsp90-dependent manner and is necessary for phosphorylation of the transcription factor IRF3 (*Liu et al., 2010*). The loss of Tom70 or its chaperone binding ability through point mutation is sufficient to produce a loss in interferon signaling (*Lin et al., 2010*; *Wei et al., 2015*; *Liu et al., 2010*). In vitro studies have shown that when Orf9b is pre-bound to Tom70, peptides of the chaperone EEVD motif have a reduced affinity (*Gao et al., 2021*). However, the pre-binding of Hsp70 to the chaperone motif does not inhibit binding of MTS-containing substrates to the C-terminal domain (*Mills et al., 2009*). Two groups have separately reported that the Orf9b:Tom70 complex formed by co-expressing the two proteins together (*Gordon et al., 2020*; *Gao et al., 2021*). Further, both reported that incubating separately purified Orf9b and Tom70 does not result in complex formation. These data suggested a model where Orf9b expressed alone purified as a very stable homodimer with no ability to form a complex when incubated with Tom70 (*Gordon et al., 2020*; *Gao et al., 2021*). Given that Tom70-binding results in a suppression of type 1 IFN, it is unclear how Orf9b homodimers (which do not directly bind to Tom70) affect viral replication.

The existence of two distinct folding conformations and oligomeric states suggests that Orf9b is in a complex equilibrium between homodimer and monomer states. Two copies of Orf9b can homodimerize, or each Orf9b monomer can bind to a single Tom70, which leads to the suppression of IFN signaling. The final equilibrium between Orf9b homodimer and Orf9b:Tom70 can be regulated by the

host cell in at least two distinct ways. First, Orf9b can be phosphorylated (*Bouhaddou et al., 2020*; *Thorne et al., 2022*) at S53 by a host kinase which prevents binding to Tom70, restoring IFN signaling. Second, crystal structures of the Orf9b homodimer show a hydrophobic central channel that runs the length of the dimer that is occupied by a lipid-like ligand (*Meier et al., 2006*; *Jin et al., 2023*). This led to an initial characterization of the Orf9b homodimer as a membrane binding protein (*Gao et al., 2021*; *Meier et al., 2006*) with subsequent work showing that lipids co-purified with recombinantly expressed Orf9b in both *E. coli* and mammalian cells (*Jin et al., 2023*). The lipid-bound homodimer is very stable and seems to not readily dissociate into monomers to bind Tom70 (*Gordon et al., 2020*; *Gao et al., 2021*). Previous efforts to isolate the lipid-free Orf9b homodimer by refolding purified protein from inclusion bodies using *E. coli* expression systems suggested that the lipid remains tightly associated during purification (*Jin et al., 2023*). Presumably, the stability afforded by lipid binding biases the equilibrium towards the homodimeric state, which does not directly bind to Tom70. However, the potential regulatory role of lipid binding to Orf9b during viral infection is unknown.

In addition, Orf9b has acquired mutations at both the protein and RNA level through several variants of concern (VOC; *Yang et al., 2023*); however, none of the coding mutations have been characterized for their effect on either affinity for Tom70 or on homodimer stability. Assigning a functional role to these mutations is complicated by the fact that Orf9b is encoded through an alternative open reading frame within the nucleocapsid (N) gene (*Rota et al., 2003*; *Marra et al., 2003*). Thus, whether these mutations have an effect on Orf9b's role in innate immune suppression or are selected due to their effect on nucleocapsid function is unknown. Mutations that regulate the expression of Orf9b have been shown to result in an increase in the expression of Orf9b at both the transcriptomic and protein levels (*Thorne et al., 2022*; *Yang et al., 2023*), which also leads to more potent suppression of IFN signaling. This result suggests that increases in Orf9b expression and coding mutations could alter the monomer:dimer equilibrium, and consequently interferon signaling.

In this work, we developed a series of ordinary differential equations (ODE) that describe the complex equilibrium between Orf9b oligomerization status and binding to Tom70. We experimentally identified model parameters for the coupled equilibrium using both surface plasmon resonance (SPR) and a fluorescent polarization peptide displacement assay that we developed. We used this model to characterize the effect of lipid-binding and mutations present in VOC on Orf9b homodimer stability. We observed that mutations to Orf9b can either slow or accelerate the time to reach equilibrium, but do not greatly affect the affinity of the interaction between Orf9b and Tom70. In contrast, we found that lipid binding strongly shifts the equilibrium towards the dimer and slows Orf9b binding to Tom70 by stabilizing the Orf9b homodimer in vitro. In summary, our model can provide biophysical insights into mutation and ligand-based shifts in the Orf9b monomer:dimer equilibrium, which has downstream implications for understanding the course of interferon signaling in coronavirus infection.

## Results

### Orf9b-derived peptides inform on the monomer:Tom70 equilibrium

The existence of two distinct structures suggests that Orf9b is in equilibrium between a dimeric and monomeric form. From this equilibrium, two copies of Orf9b homodimerize or one copy of Orf9b can bind to Tom70 (*Figure 1A*). We started developing a biophysical model by considering a monomeric intermediate state as Orf9b undergoes conformational transitions between its α-helical Tom70 binding form and its β-sheet-rich homodimer form. To constrain this model, we first measured the equilibrium between Tom70 and the Orf9b monomer binding.

We used a previously characterized peptide for Orf9b that retains the residues that form the helix to bind Tom70, but does not have the core residues needed to form the β-sheet-rich homodimer (*Gao et al., 2021*). This peptide represents approximately ⅓ of the full-length Orf9b sequence and can be used as a proxy for studying the behavior of the monomeric Orf9b intermediate. We determined the rate constants of the peptide Orf9b monomer binding to Tom70 with surface plasmon resonance (SPR) using immobilized Tom70 and the Orf9b peptide as the analyte. Global fitting of the Orf9b peptide:Tom70 binding curves resulted in a $K_D$ of 653±2 nM (*Figure 1B*). These fits were associated with kinetic rates of $k_{on} = 6.2 \times 10^4$ M$^{-1}$s$^{-1}$ and $k_{off} = 4.1 \times 10^{-2}$ s$^{-1}$; however, the fitting procedure used by our SPR setup does not generate individual error estimates for these rates. The Orf9b peptide:Tom70 association rate was much slower than the diffusion limit, which may suggest that a

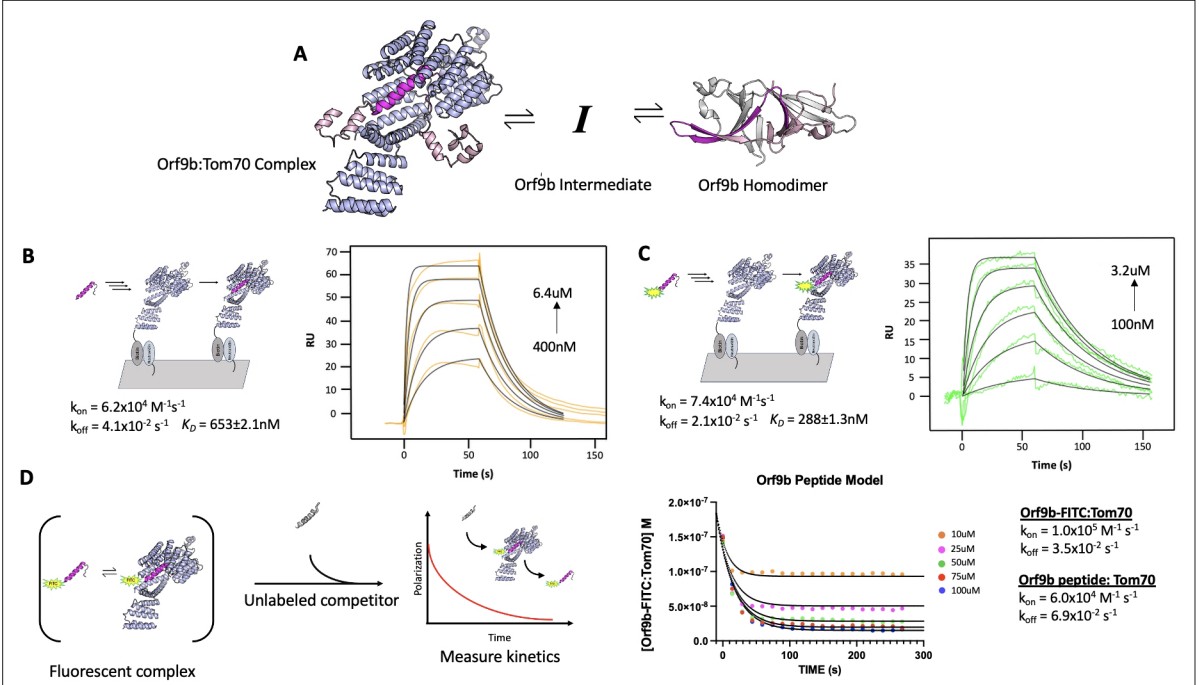

**Figure 1.** Peptides of Orf9b can be used to model the behavior of monomeric Orf9b to Tom70. (**A**) Initial model of the Orf9b-Tom70 equilibrium. (**B**) (Left) Schematic of SPR binding interaction between Tom70 and Orf9b peptide. (Right) Surface plasmon resonance sensorgram of Orf9b peptide binding to immobilized Tom70 (orange) with model fits (black) showing kinetics and dissociation constant. (**C**) (Left) Schematic of SPR binding interaction between Tom70 and Orf9b-FITC. (Right) Surface plasmon resonance sensorgram of Orf9b-FITC peptide binding to immobilized Tom70 (green) with model fits (black) showing kinetics and dissociation constant. (**D**) (Left) Diagram of Orf9b-Tom70 fluorescent polarization assay. (Right) Model ODE (solid lines) overlaid with Orf9b peptide competition kinetics (circles) and model parameters used to generate results.

The online version of this article includes the following figure supplement(s) for figure 1:

**Figure supplement 1.** Determining the KD of Orf9b peptides for Tom70.

**Figure supplement 2.** S53E phosphomimetic Orf9b peptides do not bind to Tom70.

**Figure supplement 3.** Plots of residuals from Orf9b peptide model showing effect of an increase or decrease by 10% on each model parameter.

conformational change occurs either by the peptide when adopting the helical conformation or by Tom70 to open up the C-terminal binding site.

As a complementary method for measuring Orf9b binding to Tom70, we appended a C-terminal fluorescein fluorophore (*Gao et al., 2021*) to the Orf9b peptide (Orf9b-FITC) to measure changes in fluorescence polarization (FP) upon Tom70 binding. When the probe is bound to Tom70, the slower tumbling of the Orf9b-FITC:Tom70 complex results in emission of more polarized light compared to the isolated Orf9b-FITC in solution. When the unlabeled Orf9b peptide is added to the pre-equilibrated Orf9b-FITC:Tom70 solution as a competitor, the polarization decreases. This decrease could be used to indirectly measure the time it takes for the Orf9b monomer to reach equilibrium with Tom70. First, we verified Orf9b-FITC binding to Tom70 using the same SPR format as the Orf9b peptide. Globally fitting a 1:1 Langmuir binding model to Orf9b-FITC:Tom70 resulted in a $K_D$ of 288±1 nM with kinetic rates $k_{on} = 7.4 \times 10^4$ $M^{-1}s^{-1}$ and $k_{off} = 2.1 \times 10^{-2}$ $s^{-1}$ (*Figure 1C*). Next, we used FP to orthogonally determine the probe's affinity for Tom70 by titrating Tom70 against a fixed concentration of Orf9b-FITC. Fitting with a 1:1 binding model yielded a $K_D$ of 240±7 nM (*Figure 1—figure supplement 1A*), which is in good agreement with the SPR result and suggests that the FITC moiety slightly favors binding relative to the unlabeled peptide.

To further confirm that the fluorophore strengthened the interaction with Tom70, we competed unlabeled Orf9b peptide against the pre-equilibrated Orf9b-FITC:Tom70 complex. In this mode, the affinity of the unlabeled peptide can be determined by titrating it against a fixed concentration of the pre-equilibrated Orf9b-FITC:Tom70 solution by measuring the decrease in polarization signal upon displacement of Orf9b-FITC by unlabeled Orf9b peptide. We fit a one-site competitive binding model

to this dataset, which yielded a $K_i$ of 880±5 nM which is consistent with the SPR measurement of the $K_D$ for Orf9b peptide:Tom70 (**Figure 1—figure supplement 1B**). As a negative control, we used an Orf9b peptide with an S53E phosphomimetic mutation (**Figure 1—figure supplement 2B**). S53 on Orf9b is positioned deep within the C-terminal Tom70 binding site and has been shown to undergo phosphorylation in cells infected with SARS-CoV-2 (**Bouhaddou et al., 2020**; **Bouhaddou et al., 2023**; **Figure 1—figure supplement 2A**). As expected, we did not see any decrease in FP signal confirming that the S53E mutation directly blocks Orf9b from binding Tom70 (**Figure 1—figure supplement 2C**).

With the long-term goal of expanding to a kinetic model that incorporates Orf9b dimerization, we wanted to assess the agreement with the rates measured by SPR and the competition experiment. We used Berkeley Madonna to construct a series of differential equations that describe the time evolution of the binding of unlabeled peptide to Tom70 and concomitant displacement of Orf9b-FITC. The model was fit to the Orf9b-FITC:Tom70 concentration time series data, which decreases over time as the unlabeled Orf9b peptide saturates Tom70. We converted the polarization values from our assay to concentrations of the Orf9b-FITC:Tom70 fluorescent complex (see Materials and methods) and derived the following differential equations, which were used to model the experimental data (where all bracketed values are concentrations):

1. d[Orf9b-FITC]/dt=-k1[Orf9b-FITC][Tom70]+k2[Orf9b-FITC:Tom70]
2. d[Orf9b peptide:Tom70]/dt = k5[Orf9b peptide][Tom70]-k6[Orf9b peptide:Tom70]
3. d[Orf9b peptide]/dt=-k5[Orf9b peptide][Tom70]+k6[Orf9b peptide:Tom70]
4. d[Tom70]/dt=-k1[Orf9b-FITC][Tom70]-k5[Orf9bpeptide][Tom70]+k6[Orf9b peptide:Tom70]+k2[Orf9b-FITC:Tom70]

We used the on and off rates determined from the SPR experiments as initial model parameters and then manually adjusted the rates to achieve the closest agreement with the 100 µM peptide condition. With the parameter set obtained from the 100 µM condition, we then held all parameters fixed and simply changed the peptide concentrations in the model to fit the remaining conditions by hand. We note that this process saw the model parameter values change between 3% at the lowest end up to 70% at the highest end from the experimentally derived values but remained within an order of magnitude of the experimental SPR values. We speculate that this arises due to the differences in experimental setup between SPR and FP-based methods of measuring kinetics. The solutions generated from the model closely match the experimental data (**Figure 1D**), suggesting that the model parameters used describe our other conditions where only the concentration of the peptide changes. While methods exist for predicting confidence intervals (CIs) on parameter estimates generated from ODE analysis, such as bootstrapping (**Joshi et al., 2006**), they are computationally demanding and often only produce questionable confidence estimates for ODEs that are often difficult to interpret (**Kreutz et al., 2012**). As an alternative to attempting to place CIs on the parameters, we performed sensitivity analysis to determine which parameters the model was most sensitive to (see methods and **Figure 1—figure supplement 3**). Additionally, we note that the model parameters were derived from the fit of only one concentration (100 uM), but fit the other concentrations equally well. We observed that the model parameter that was most sensitive to change was the rate of Orf9b-FITC:Tom70 dissociation when subjected to a 10% increase or decrease, whereas all other model parameters showed no sensitivity to change (**Figure 1—figure supplement 3**). This analysis, coupled with inspection of the residuals, provides strong support for the accuracy of the model and the fitted parameters.

## A refolding procedure to generate lipid-free Orf9b homodimers

Crystal structures of the Orf9b homodimer for both SARS-CoV (**Meier et al., 2006**, PDB 2CME) and SARS-CoV-2 (**Jin et al., 2023**, 7YE7, 7YE8, and unpublished structure 6Z4U) show a central channel that is formed by the two copies of the Orf9b homodimer. Within the central channel, there is additional electron density that has been modeled as either a linear alkane or polyethylene glycol (PEG). Presumably, an alkane could bind in the expression organism and persist during the purification, whereas the PEG would bind from crystallization conditions that include it as a precipitant. However, additional structures of the Orf9b homodimer show the same density in crystallization conditions that lack PEGs (**Jin et al., 2023**, PDB: 7YE7 and 7YE8). This suggests that the additional density within the Orf9b homodimer central channel does not arise from the crystallization conditions, but rather from the expressing organisms and is retained during the purification of the protein. Indeed, mass spectrometry suggests that recombinantly purified Orf9b binds lipids (**Jin et al., 2023**). We hypothesized

that the binding of lipids to the Orf9b homodimer could stabilize the homodimer and prevent dissociation into monomeric Orf9b. This stabilization could explain previous observations that separately purified Orf9b homodimer and Tom70 do not readily form an Orf9b:Tom70 complex (*Gordon et al., 2020*).

To test this idea, we attempted to isolate the lipid-free Orf9b homodimer by performing a denaturing and refolding purification process (*Figure 2A*). We note that our approach has key differences from the previous efforts by Jin et al., who refolded recombinantly expressed Orf9b isolated from inclusion bodies in *E. coli* expression systems. Their crystal structure (PDB ID: 7YE8) still shows clear density within the hydrophobic central channel, supporting the placement of an 8-carbon alkane. Unlike Jin et al., we purified Orf9b in the presence of Guanidine HCl and detergent so that the Orf9b isolated by nickel column chromatography was completely denatured. This step ensures the loss of co-purifying lipids prior to refolding. The SEC elution profile and retention volume of refolded Orf9b directly overlapped with natively folded homodimeric Orf9b, with the early eluting peaks corresponding to either a chaperone-bound species (natively folded) or misfolded protein (refolded) as

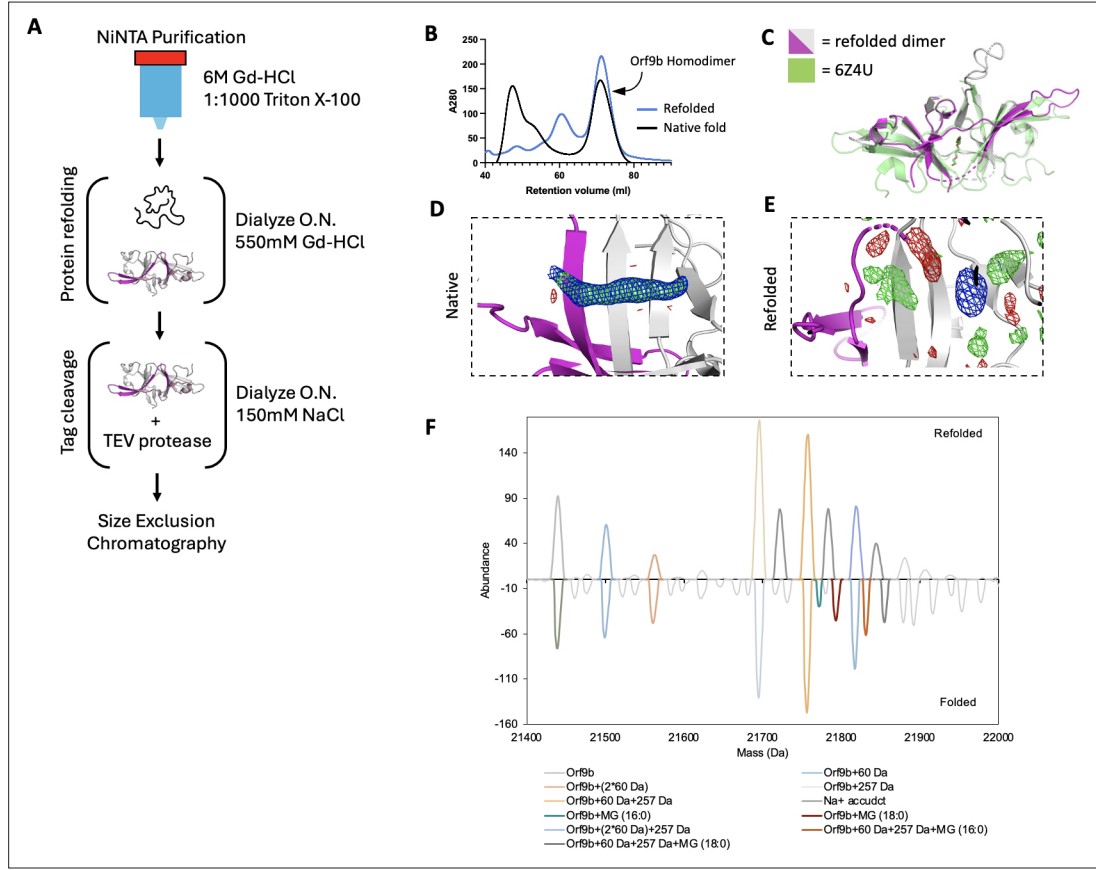

**Figure 2.** The Orf9b homodimer can be refolded to eliminate a co-purifying lipid. (**A**) Schematic of Orf9b denaturing purification, refolding, and tag-cleavage process. (**B**) Size exclusion chromatography chromatogram overlay of WT natively folded (black) and WT refolded (blue) Orf9b homodimer. Both proteins purify as a dimer and have identical retention volumes indicating high recovery of refolded Orf9b homodimer. (**C**) Structural overlay of WT Orf9b homodimer (green-PDB 6Z4U) and WT refolded Orf9b homodimer (magenta/gray-PDB 9MZB). (**D**) Zoomed-in view of the natively folded Orf9b homodimer central channel. 2Fo-Fc (contoured to 1σ) and Fo-Fc (+/-3σ) density maps show continuous density that supports the placement of a ligand in the dimer central channel. (**E**) Zoomed-in view of the refolded Orf9b homodimer central channel: 2Fo-Fc (1σ) and Fo-Fc (+/-3σ) density maps show diffuse peaks within the central channel of the refolded Orf9b homodimer. (**F**) Deconvolved native mass spectra of refolded and natively folded Orf9b homodimers in negative-polarity mode. Lipid-bound homodimers are observed only in the natively folded sample. Both preparations display +60 Da and +257 Da adducts of unknown origin.

The online version of this article includes the following figure supplement(s) for figure 2:

**Figure supplement 1.** Tandem mass spectrometry reveals the identities of possible lipids bound to the Orf9b homodimer in different expression systems.

**Figure supplement 2.** Polder maps support the absence of lipids bound to the refolded Orf9b homodimer.

judged by SDS-PAGE (*Figure 2B*). Together, the overlap in elution peaks corresponding to the folded homodimer suggested a high recovery of the homodimer from the refolding conditions.

## Crystal structure of refolded Orf9b homodimer reveals that lipid stabilizes the dimer, but is not essential for its formation

The electron density assigned to the lipid molecule is in a central channel between the two chains of the Orf9b homodimer. This channel is approximately 22 Å in length and lined with primarily hydrophobic residues. The potential for complementary hydrophobic interactions with the lipid along the channel led us to hypothesize that such interactions could stabilize the Orf9b homodimer, biasing the Orf9b monomer:dimer equilibrium. To further confirm that the refolding process resulted in an Orf9b homodimer, we crystallized the refolded homodimer and determined a 2.8 Å structure (*Supplementary file 1*). We performed molecular replacement using the WT reference structure and observed that the refolded homodimer structure closely aligned with the lipid-bound reference structure, which shows that the homodimer fold can be recovered after denaturing (*Figure 2C*). Aligning the structure of the Orf9b homodimer (PDB 6Z4U) with our structure of the refolded Orf9b homodimer (9N55) in Pymol resulted in an RMSD of 1.1 Å. Further, we also searched our structures of the refolded Orf9b homodimer on the Dali server against the existing structures of the lipid-bound Orf9b homodimer, which yielded a Z-score of 2.2, which shows good correspondence between the structures. Compared to the reference PDB structure, we were also able to resolve one of two disordered loops, which is likely due to the refolded homodimer crystallizing in a different crystal form and space group than the reference structure.

As we hypothesized, when we inspected the central channel of the refolded homodimer, we did not observe significant electron density in the central hydrophobic channel to support the placement of the lipid but did observe some diffuse density peaks in both the Fo-Fc and 2Fo-Fc maps (*Figure 2E*). Given that our crystallization conditions contained glycine and the SEC buffer contained glycerol, we attempted to model different buffer components into the density peaks. We also crystallized natively purified Orf9b to generate electron density maps that we could compare to the refolded homodimer. The natively purified Orf9b homodimer had clear density in the central channel just like the existing structures which supported modeling of an 8-carbon alkane (*Figure 2D*, *Supplementary file 1*). We generated Polder maps (*Liebschner et al., 2017*) for both the natively folded lipid-bound structure as well as the refolded structure with either glycine or an 8-carbon alkane (representing the lipid) model to confirm that the diffuse electron density in the refolded homodimer was not due to lipid density being obscured by the bulk solvent mask. For the natively folded structure, the Polder maps clearly supported the presence of an 8-carbon alkane (*Figure 2—figure supplement 2A*); however, for the refolded structure, Polder maps did not support the placement of the 8-carbon alkane (*Figure 2—figure supplement 2B*). We next modeled glycine into one of the density peaks and noted a good fit with little residual difference density (*Figure 2—figure supplement 2C*). Therefore, we conclude that the remaining density peaks in the channel are likely attributed to noise or crystallization buffer components. These results also suggest that the lipid binding is not necessary for Orf9b dimerization; however, we cannot definitively rule out the possibility that the presence of lipids can also induce Orf9b dimerization.

We speculate that the reason why our structure of the refolded Orf9b homodimer lacks density to support lipid placement, but the refolded structure of Jin et al. has density consistent with a lipid is due to differences in methods for refolding the homodimer. We differed from Jin et al. by purifying in the presence of strong denaturant to ensure that co-purifying lipids could not remain bound, whereas Jin et al. purified from inclusion bodies. Purifying from inclusion bodies may not result in fully unfolded protein and may contain enough lipid-bound homodimers to produce clear lipid density when resolved by X-ray crystallography.

## Native mass spectrometry reveals lipid species bound to Orf9b homodimers

Next, we utilized native mass spectrometry to confirm lipid binding to Orf9b homodimer. The deconvoluted native mass spectra revealed that the natively folded Orf9b exhibits unique peaks at +332 and +358 Da, which are absent in the refolded sample (*Figure 2F*). To characterize the bound lipids, we performed a lipid extraction on the natively folded Orf9b homodimer purified from *E.*

*coli* and analyzed the resulting lipids by mass spectrometry. The MS and MS/MS of the extracted lipids confirmed that the bound lipids are 1-palmitoyl-sn-glycerol (MG 16:0) or 1-stearoyl-sn-glycerol (MG 18:0; *Figure 2—figure supplement 2A*). These assignments are consistent with the observed mass shifts from the native mass spectrometry data and previous results (*Jin et al., 2023*). To test whether these co-purifying lipids were bound to Orf9b purified from more physiologically relevant cell types, we expressed and purified WT Orf9b from mammalian cells. Similarly to Jin et al., our sample displayed similar MS patterns to the *E. coli*-derived sample and the assignments were consistent with 1-palmitoyl-sn-glycerol or 1-stearoyl-sn-glycerol (*Figure 2—figure supplement 2B and C*). Collectively, these results suggest that Orf9b homodimer lipid-binding is not limited to the cell types it is expressed in. We also conclude that the lipid can unbind when Orf9b is completely unfolded and that Orf9b can form homodimers in the absence of lipid. Going forward, we will refer to the refolded Orf9b homodimer as the apo-homodimer and the natively folded Orf9b homodimer as the lipid-bound homodimer.

## Lipid binding shifts the Orf9b dimer:monomer equilibrium towards the dimer

To test whether there was a difference in homodimer stability between apo and lipid-bound Orf9b homodimers, we first monitored the concentration dependence of SEC elution profiles. Performing SEC on diluted apo-Orf9b homodimer sample produced two peaks on the chromatogram: the first peak corresponded to the homodimer and the second peak corresponded to the monomer (*Figure 3*). For the Orf9b homodimer, the retention volume was consistent with molecular weight standards based on the expected molecular weight of the homodimer (~21 kDa) and the standard (~29 kDa). In the case of the Orf9b monomer, although we would expect the retention volume of the monomer (~10.6 kDA) to be between the molecular weight standards of 13.4 kDa and 6.5 kDa, the greater retention volume could be explained by non-specific hydrophobic interactions between the monomeric Orf9b and the column. We confirmed that both peaks were Orf9b by SDS-PAGE (*Figure 3B*). In contrast, the lipid-bound homodimer remains homodimeric at multiple concentrations (*Figure 3C*). We next sought to determine the dissociation constant of the apo-homodimer by monitoring the concentration dependence of the SEC elution profiles. As the concentration of apo-Orf9b homodimer decreased, the ratio between the area under the dimer elution peak and the monomer elution peak decreased (*Figure 3D*). We fit the total concentration of Orf9b versus the fraction of Orf9b in the homodimer form with a non-linear regression and determined a $K_D$ of 1.2±0.1 uM (*Figure 3E*). Although we were limited by the sensitivity of SEC at low micromolar concentrations, the stark difference between apo (*Figure 3E*) and lipid-bound (*Figure 3C*) Orf9b at low micromolar concentrations suggests that the binding of the lipid tightly stabilizes the Orf9b homodimer, preventing dissociation and monomer formation.

We also confirmed that apo-Orf9b homodimer could be used to produce the Orf9b:Tom70 complex when incubated with Tom70. We incubated separately purified Tom70 and refolded (apo) Orf9b homodimer together and then ran them over an SEC column. We ran Tom70 alone followed by the Orf9b:Tom70 incubation and saw a left-shifted retention volume relative to Tom70 alone with a right-handed shoulder indicating a mix between Orf9b:Tom70 and Tom70 alone (*Figure 3—figure supplement 1A*). SDS-PAGE gel of the fractions showed that the first fractions contained both Orf9b and Tom70, followed by a fraction containing mostly Tom70 alone and finally Orf9b alone (*Figure 3—figure supplement 1B*). This suggests that the formation of the Orf9b:Tom70 complex is strongly influenced by the presence (or absence) of the lipid when starting with the Orf9b homodimer.

## Lipid binding to the Orf9b homodimer substantially slows Orf9b binding to Tom70

To quantify the stabilization afforded by lipid binding, we returned to our FP assay (*Figure 1D*). Our hypothesis was that the elimination of the lipid would allow Orf9b homodimers to dissociate into monomers more rapidly and that this could be monitored by binding to Tom70 more readily. We used the FP assay with lipid-bound and apo-Orf9b homodimer as the competitors to pre-equilibrated Orf9b-FITC:Tom70 fluorescent complexes. The apo-Orf9b homodimer equilibrated rapidly (500–1000 ss) with Orf9b-FITC:Tom70. These conditions showed concentration-dependent behavior, consistent with the coupled equilibria (*Figure 4Aii*): at high total Orf9b concentration,

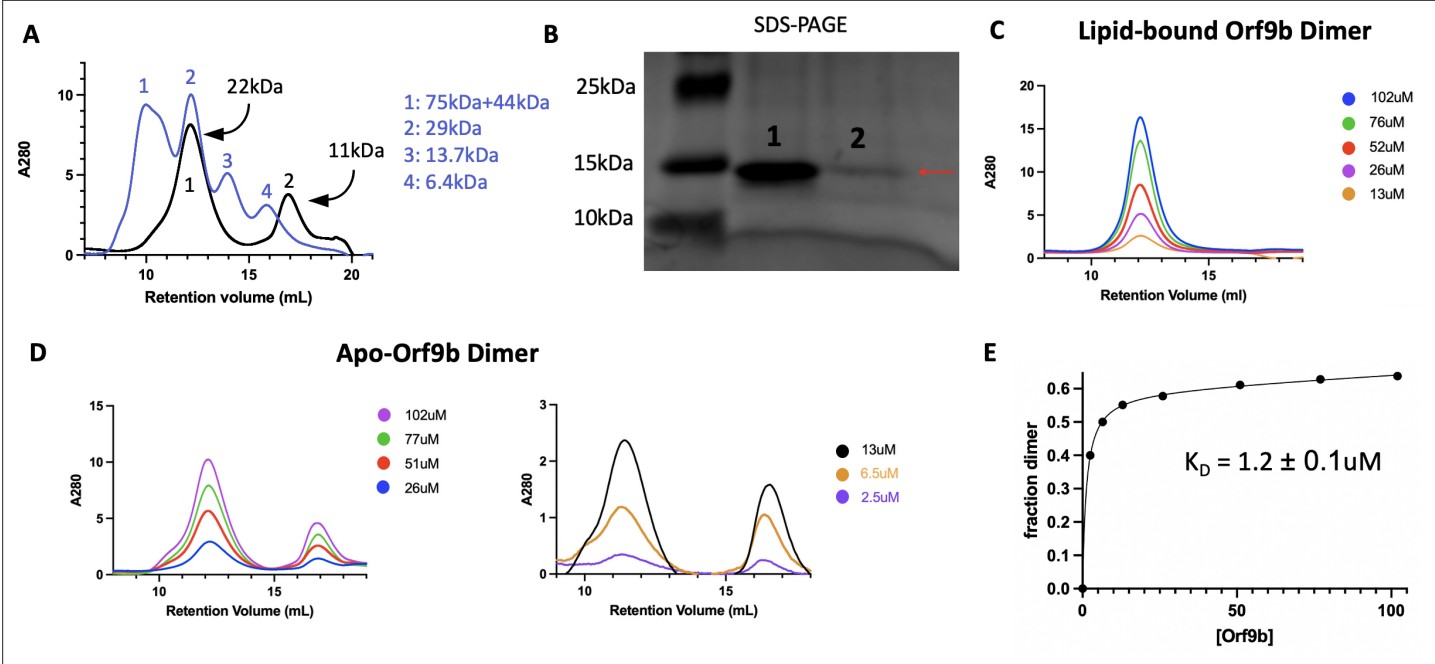

**Figure 3.** Serial dilution allows for estimation of Orf9b homodimer dissociation constant. (**A**) Size exclusion chromatography overlay of Orf9b homodimer (black) with molecular weight standards (blue). Apo-Orf9b forms two distinct peaks corresponding to the dimer (peak 1) and the monomer (peak 2). (**B**) Coomassie stain of SDS-PAGE gel from peaks 1 and 2 (from **A**) showing Orf9b is present in each peak. (**C**) Size exclusion chromatography chromatogram of serially diluted WT lipid-bound Orf9b homodimer showing no monomer species present. (**D**) Size exclusion chromatography chromatogram of serially diluted WT apo-Orf9b showing both homodimer and monomer peaks. Running molecular weight standards further supported our observation that the first peak was the Orf9b homodimer due to its close overlap with a 29 kDa standard; however, the monomeric peak eluted after a 6.5 kDa standard rather than between the 13.7 kDa and 6.5 kDA standards as we would expect. We hypothesize that this discrepancy is due to two possibilities: hydrophobic interactions between the monomer and the column could increase the retention volume or monomeric Orf9b has a smaller hydrodynamic radius than expected. (**E**) Nonlinear regression analysis of the fraction of Orf9b in the homodimer versus the total concentration of Orf9b (from **D**) injected over the size exclusion column yields a $K_D$ of 1.2±0.1 uM.

The online version of this article includes the following source data and figure supplement(s) for figure 3:

**Source data 1.** Original file containing SDS-PAGE analysis for *Figure 3B*.

**Source data 2.** PDF file containing original SDS-PAGE for *Figure 3B* with indicated bands labels.

**Figure supplement 1.** Refolded Orf9b binds to Tom70 and co-elutes as a complex by size exclusion chromatography.

**Figure supplement 1—source data 1.** Raw SDS-PAGE of Orf9b homodimer and monomer SEC peaks.

**Figure supplement 1—source data 2.** PDF of raw SDS-PAGE of Orf9b homodimer and monomer SEC peaks with molecular weights and relevant bands marked.

the displacement curves resembled a single exponential; however, at lower concentrations, a more pronounced sigmoidal behavior appeared. We next performed the competition experiment with lipid-bound Orf9b homodimer. Strikingly, we observed that the lipid-bound Orf9b homodimer was much slower to bind to Tom70 (as monitored by the decrease in fluorescent polarization) than for the apo protein, with equilibrium reached only after several hours of incubation (*Figure 4Ai*). As a negative control, we purified homodimeric Orf9b S53E. As we observed for the control Orf9b-S53E peptide, experiments with purified homodimeric Orf9b S53E did not cause any decrease in FP signal (*Figure 4—figure supplement 1*). These results support our hypothesis that the decrease in FP signal over time is driven by the Orf9b homodimer dissociating into monomers and the monomers binding to Tom70 to displace the Orf9b-FITC probe with the difference in equilibration time driven by the lipid slowing the dissociation of the homodimer. We next sought to incorporate the rates from our SPR data into a model that describes how this more complex equilibrium is reached in our FP assay when lipid is either present or absent.

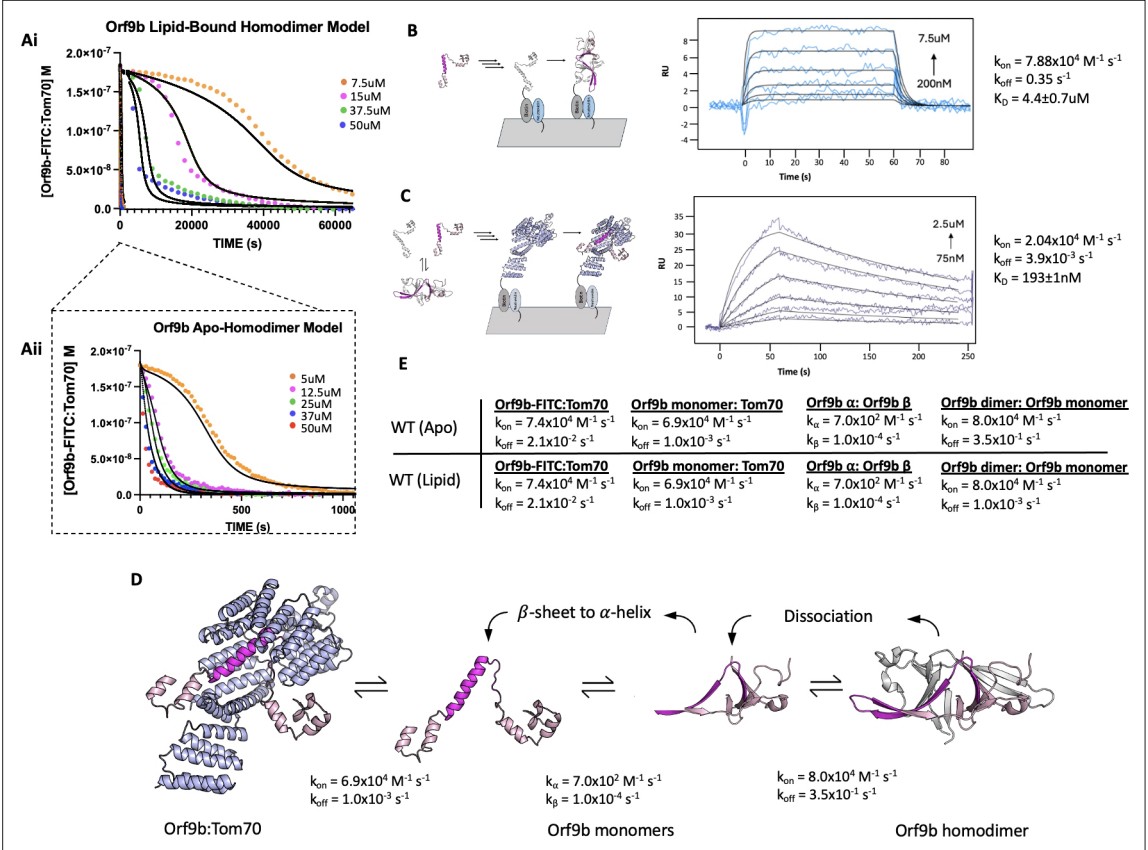

**Figure 4.** Modeling the effect of lipid-binding on Orf9b-Tom70 equilibrium using SPR and FP-based assay. (**A**) I. Mathematical model overlay to FP competition kinetic assay using the lipid-bound Orf9b homodimer as the competitor. II. Mathematical model overlay to FP competition kinetic assay using the apo-Orf9b homodimer as the competitor. (**B**) (Left) Schematic diagram of monomeric Orf9b binding to immobilized monomeric Orf9b to determine dimerization rate constants. (Right) Surface plasmon resonance sensorgram of full-length Orf9b binding to full-length Orf9b monomer and their interaction kinetics. Experimental binding curves (blue) were globally fit (black) using a 1:1 binding model. (**C**) (Left) Schematic of full-length apo-Orf9b homodimer in exchange with Orf9b monomers. Monomeric Orf9b binds to immobilized Tom70 to determine rate constants of full-length Orf9b binding to Tom70. (Right) Surface plasmon resonance sensorgram of refolded apo-Orf9b binding to Tom70 and their interaction kinetics. Experimental binding curves (dark blue) were globally fit (black) using a 1:1 binding model. (**D**) Proposed kinetic model of Orf9b-Tom70 equilibrium. Orf9b homodimer dissociates into Orf9b monomers which undergo a conformational change from β-sheet to α-helix. α-helical Orf9b binds to Tom70. (**E**) Table of model parameters used to model ODEs to both apo and lipid-bound Orf9b homodimer competition binding kinetics.

The online version of this article includes the following figure supplement(s) for figure 4:

**Figure supplement 1.** Orf9b S53E refolded homodimer FP kinetic assay results showing that the phosphomimetic mutation does not bind to Tom70.

**Figure supplement 2.** Predicted iMTS regions outside of the structurally resolved portions of Orf9b bound to Tom70 do not bind to Tom70.

**Figure supplement 3.** Comparison of kinetic model 1 and 2 in describing experimental results from the kinetic binding assay.

**Figure supplement 4.** Plots of model behavior showing the effect of changes to alpha-beta and beta-alpha monomer interconversion rates compared to experimental values.

**Figure supplement 5.** Plot of residuals showing the effect of increasing or decreasing individual model parameters 10% compared to the reported values.

## SPR measurements for Orf9b dimerization and Tom70 binding constrain models of the coupled equilibria

To model the competition FP data, we can leverage our prior SPR experiments, which report on the $k_{on}/k_{off}$ for either Orf9b monomer or Orf9b-FITC interacting with Tom70. However, we also needed initial estimates of the rate constants for Orf9b oligomerization to constrain the model. To derive this rate, we isolated the monomeric fraction of biotinylated Orf9b and immobilized it on an SPR chip surface. We used a low immobilization signal of 15 RU to minimize the chance of biotinylated Orf9b monomers cross-dimerizing on the chip. To obtain the unlabeled monomeric Orf9b, purified

WT Orf9b homodimer was denatured in 6 M Guanidine HCl overnight followed by injection over a sizing column pre-equilibrated in the SPR running buffer. This procedure allowed us to isolate the monomeric fraction as well as desalt the sample into the SPR running buffer to minimize buffer mismatch artifacts. Global fitting of the binding curves using a 1:1 Langmuir binding model yielded a $K_D$ of 4.4±0.7 µM which is within the order of magnitude of the dissociation constant calculated by dilution (1.2±0.1 µM; *Figure 3E*). The kinetic rates were $k_{on}$ = 7.88 × $10^4$ $M^{-1}s^{-1}$ and $k_{off}$ = 3.5 × $10^{-1}$ $s^{-1}$ (*Figure 4B*). The maximum response during binding was approximately 8 RU, which was close to the theoretical maximum response of 11 RU for a 1:1 interaction. The slow association rate showed that binding was not diffusion limited. This slower rate may be due to Orf9b undergoing conformational changes prior to dimerization on the surface.

Next, we immobilized Tom70 to the chip surface, as in the Orf9b peptide experiments, and injected apo-Orf9b. Our experimental measurement of the Orf9b homodimer dissociation constant was in the low single-digit micromolar; therefore, under the concentrations that we tested for the Tom70-immobilized SPR experiment, the oligomeric state of apo-Orf9b should be largely monomeric. When we performed global fitting of our Orf9b-Tom70 sensograms, we found that a 1:1 interaction showed a good fit, yielding a $K_D$ for Orf9b:Tom70 of 190±1 nM. The dissociation rate was approximately 10-fold slower than the peptide and the association rate was also approximately 3 times slower than estimated for the Orf9b peptide (*Figure 4C*). To explain these altered kinetics, we considered whether Orf9b can bind to more than one copy of Tom70 via the MTS motif. This idea was motivated by AlphaFold predictions of monomeric WT Orf9, which show two helices: the first encompassing residues 11–28 and the second encompassing residues 44–70 (*Figure 4—figure supplement 2A*). While the 44-70aa helix is resolved in both crystal and cryo-EM structures as bound to Tom70, the 11-28aa helix could form an MTS that may be recognized by a second copy of Tom70. To test this idea, we generated a peptide for 11-28aa, but did not observe any signal for binding by the FP assay or SPR (*Figure 4—figure supplement 2B and C*). With only single-site binding, we speculate that the difference between the Orf9b peptide and full-length kinetics is due to the Orf9b peptide truncating several residues that are responsible for binding to Tom70 that influence both the association and dissociation pathways.

## Incorporating conformational change into the kinetic model of Orf9b oligomerization and Tom70 binding

Our initial models, which incorporated only a simple intermediate as Orf9b monomers and Orf9b homodimers interconvert (*Figure 1A*), did not accurately capture the behavior observed in the FP assay (*Figure 4—figure supplement 3*). We revised our model to now explicitly include model parameters that describe the rate of interconversion and conformational change between β-sheet and α-helical monomers (*Figure 4D*). The differential equations for the model were the following:

1. d[Orf9b-FITC]/dt = -k1[Orf9b-FITC][Tom70]+k2[Orf9b-FITC:Tom70]
2. d[Orf9b β-monomer]/dt = 2(-k3[Orf9b β-monomer]^2+k4[Orf9b dimer])-kα[Orf9b β-monomer]^2+kβ[Orf9b α-monomer]
3. d[Orf9b dimer]/dt = k3[Orf9b β-monomer]^2-k4[Orf9b dimer]
4. d[Tom70]/dt=-k1[Orf9b-FITC][Tom70]-k5[Orf9b α-monomer][T]+k6[Orf9b α-monomer:Tom70]+k2[Orf9b-FITC:Tom70]
5. d[Orf9b α-monomer:Tom70]/dt = k5[Orf9b α-monomer][Tom70]-k6[Orf9b α-monomer:Tom70]
6. d[Orf9b α-monomer]/dt = -k5[Orf9b α monomer][Tom70]-kβ[Orf9b α-monomer]+k6[Orf9b α-monomer:Tom70]+kα[Orf9b β-monomer]^2

The model parameters $k_a$ and $k_B$ describe the rate of interchange between the β-sheet and α-helix monomer conformations. These parameters must be estimated by modeling because our assays do not allow us to directly measure the folding rates between these conformations. To identify these values, we performed a scan of $k_a$ and $k_B$ values that yielded the best agreement between the model and the experimental conditions (*Figure 4—figure supplement 4*). We tested different models that incorporated the interconversion between β-sheet to α-helix conformations by considering models that described a conformational change in the homodimer leading to Tom70 binding rather than monomers. None of these models adequately described our experimental results; therefore, we continued developing our model as outlined in *Figure 4D*. We found that this model adequately

described the changes in kinetic behavior for different concentrations of Orf9b, assuming the conversion of helix to β-sheet depended linearly on the concentration of Orf9b α-monomer with a first-order rate constant $k_B$. We initially tested keeping the rate constant $k_a$ first order just like $k_B$ which did yield the sigmoidal behavior we observed in the 5 µM apo-homodimer condition, but extending this analysis to additional concentrations tested resulted in substantial overestimation compared to our experimental results when holding $k_B$ at a constant value throughout. We found that when the β-sheet to α-helix rate ($k_a$) was made a second-order rate constant, we were able to hold the rate constant across all concentrations tested, suggesting a non-linearity in the monomer β-sheet concentration. One potential biophysical explanation for the order of these rate constants is that in the monomeric form, Orf9b is more likely to adopt either a disordered or helical conformation than the β-sheet conformation we see in the homodimer. The formation of the helical conformation may either pattern neighboring monomers to adopt the α-helical conformation and produce helix:helix dimers or the formation of monomers may destabilize the homodimer through multivalent interactions. While we did not directly test for this, we note examples of proteins exhibiting high structural plasticity that are capable of adopting different conformations mediated by cooperative mechanisms have been reported previously, such as in the case of RfaH and Lymphotactic (*Tuinstra et al., 2008*; *Zuber et al., 2022*). This revised model also allows us to use all of our measured rate constants and only requires the introduction of two unknown rate constants ($k_a$ and $k_B$) that can be determined by fitting.

We began by modeling the apo-Orf9b homodimer competition experiments (*Figure 4Aii*). We used the SPR-derived rate constants for both Orf9b-FITC:Tom70, refolded Orf9b:Tom70, and Orf9b dimerization as our model parameters. Given that $k_a$ and $k_B$ were not experimentally determined, we used our model to identify values that achieved the closest agreement between our model and experimental results when modeling the lowest concentration and then held those values fixed for subsequent concentrations. We found that modeling the initial population of Orf9b as entirely homodimeric achieved a good agreement with our experimental results. Our model was sensitive to the change in kinetic profile as the concentration of Orf9b decreased, resulting in the sigmoidal kinetic profile observed in the 5 µM condition and suggesting that the dominant behavior of apo-Orf9b is as the homodimer rather than a mix between homodimer and monomers. We feel this decision is justified both by the model's correspondence with the experimental data and the fact that the concentrations of apo-homodimer used are several-fold higher than the experimentally derived range of $K_D$ values. Further, we repeated the sensitivity analysis described previously for the peptide model and also considered the sensitivity of model parameters by inspecting each individually (*Figure 4—figure supplement 5*). Inspection of the residuals from the 5 µM apo-Orf9b homodimer time course showed clear patterns when individual model parameters were subjected to a 10% increase or decrease from the reported values. While our proposed model qualitatively describes the concentration-dependent change in kinetic behavior, the residual plots may suggest that additional binding reactions may also be occurring that are not captured by our model. We found that when examining the RMSD's of the lowest concentration of 5 µM, the model was most sensitive to changes in three parameters: the rate of homodimer association and dissociation and the conversion from β to α-monomers. However, at the highest concentrations (50 µM), the model was no longer sensitive to these parameters as it was most sensitive to the rate of Orf9b-FITC dissociation from Tom70, which was the rate-limiting step. Therefore, under low concentrations of Orf9b homodimer, binding to Tom70 is limited by the rate of homodimer association and dissociation as well as the conversion of Orf9b monomers to the α-helical conformation. Under high homodimer conditions, the limiting step becomes the rate at which Orf9b-FITC dissociates from Tom70.

Next, we modeled the lipid-bound Orf9b homodimer data (*Figure 4Ai*). We modeled the initial conditions prior to the addition of the fluorescent complex as entirely Orf9b homodimer with no monomeric species present, just like with the refolded apo-homodimer. To simplify our model for lipid-bound Orf9b homodimer, we did not explicitly include the lipid binding kinetics into our model and hypothesized that the model parameter that would be most sensitive to lipid binding would be the rate of homodimer dissociation into monomers. We used our previously implemented model parameters for the refolded homodimer datasets and initially modeled one experimental condition and then used those optimal parameter values to compare against the remaining data sets. We observed that the model was sensitive to the parameters describing the conversion from β-monomer to α-monomer and the homodimer dissociation rate, as we saw for the apo-homodimer dataset. We assumed that

lipid binding to the homodimer would have no effect on the rate of conversion from β to α-monomer, and we hypothesized that the model parameter that would be sensitized to lipid binding would be the rate of homodimer dissociation.

We found that in order for the model to pick up the correct behavior exhibited by the experimental data, it required an approximately 100-fold decrease to the rate of homodimer dissociation over all concentrations tested. Further, we also note that we observed sensitivity in the rate of Orf9b-FITC dissociation under high concentration of homodimer. Therefore, both model parameters are essential to the model across all concentration series with the rate of homodimer dissociation being the most sensitive parameter for the lipid-bound data.

We found that the lipid-bound homodimer dataset could be modeled by lowering the homodimer dissociation rate to $1.0 \times 10^{-3}$ s$^{-1}$, which is nearly 2 orders of magnitude from the SPR derived value for refolded Orf9b of $3.5 \times 10^{-1}$ s$^{-1}$ (*Figure 4E*). This brought the model into close alignment with the experimental data while all other parameters remained the same as those used in the refolded homodimer model. These model parameters resulted in a good agreement with the experimental data over different concentrations of lipid-bound homodimer. Importantly, this change in dissociation rate also explains the initial slow rate of binding to Tom70 at lower concentrations of Orf9b homodimer and the long equilibration time. Based on the fitted association and dissociation rates for the lipid-bound homodimer, our model suggests that the lipid-bound dissociation constant for the Orf9b homodimer drops to ~13 nM, which is ~100-fold stronger than the experimentally determined dissociation constant of 1.2–4.4 µM for the refolded homodimer (*Figure 4E*). Further, our model shows that lipid binding only acts on the rate of homodimer dissociation but does not affect the microscopic rate constant of Orf9b binding to Tom70 in our assay.

To explore the consequences of altering the different microscopic steps fit by our model in cellular contexts, we were next interested in testing mutations that biased the equilibrium towards either the Orf9b monomer or homodimer. We investigated the homodimer interface and observed that residues 91–97 of the C-terminal tail on chain A form backbone hydrogen bonds with residues 50–56 of chain B (*Figure 5A*). Our choice to truncate the 7 C-terminal amino acids was driven by the structure of the Orf9b homodimer which indicated that the 7 C-terminal residues form ⅔ of the homodimer interface; therefore, truncating them would compromise the ability for Orf9b to form stable homodimers and impede lipid binding. Due to the homodimer being symmetric, truncating residues on one chain would be mirrored on the opposite chain. We purified truncated Orf9b and observed a peak that eluted at 70 mL (the same retention volume as the homodimer) followed by a large peak near the end of the column's volume, which we attribute to the monomeric species (*Figure 5—figure supplement 1A*). We isolated the peak that corresponded to the homodimer and used it in the FP kinetic assay. We observed fast binding kinetics that resembled our results using the Orf9b peptide, which is an obligate monomer (*Figure 1E*) and suggested that the truncation destabilizes the Orf9b homodimer.

We next applied our kinetic model to describe how the truncation drives the equilibrium towards Tom70 binding relative to WT Orf9b. As with the apo and lipid-bound homodimer results, we initially focused on modeling our FP results by focusing on the dissociation rate as the model parameter most sensitive while keeping all other model parameters unchanged. The fast kinetics observed suggested to us that the truncation resulted in the Orf9b homodimer readily dissociating into monomers prior to the addition of the fluorescent complex. We believed that this was reasonable given that monomeric species formed a major peak in our SEC chromatogram (*Figure 5—figure supplement 1A*). When we ran the model assuming that the truncated Orf9b was entirely homodimeric to begin with (while increasing the homodimer dissociation rate), we saw poor agreement with our experimental results. We resorted to using the same model that we used for fitting the Orf9b peptide data (*Figure 1E*), which treats the system as a simple equilibrium between Orf9b-FITC:Tom70 and Orf9b monomers and produced good agreement between the model and experimental results (*Figure 5B*). We interpreted this to mean that the truncation to Orf9b results in a homodimer that is substantially less stable than the WT homodimer and that our experimental results are capturing a largely monomeric binding behavior to Tom70 without influence of the homodimer. Therefore, we interpret these results to indicate that truncating the 7 C-terminal amino acids compromises Orf9b ability to remain a stable homodimer and drives the equilibrium strongly towards monomeric Orf9b, rendering it a good reagent for investigating the role of the monomer in cellular models. Importantly, this data showed no influence of the lipid to the observed kinetics

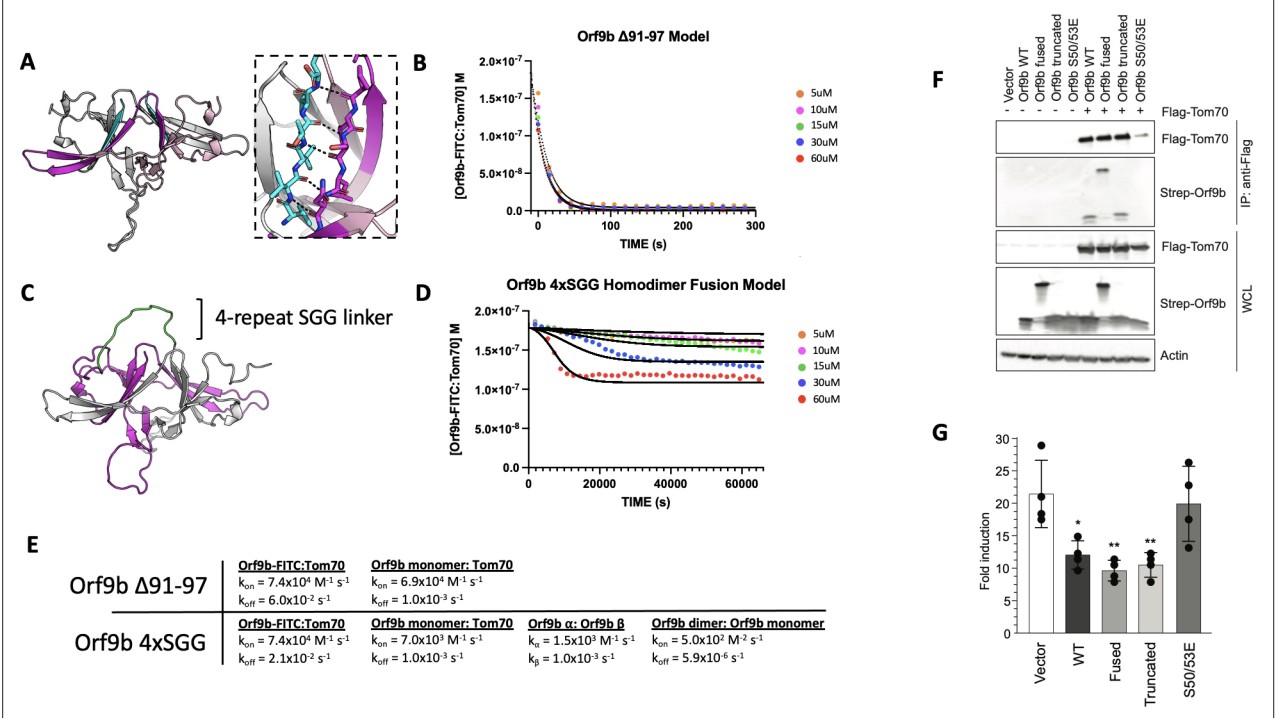

**Figure 5.** Truncated and fused Orf9b constructs can destabilize or stabilize the homodimer to modulate Tom70 binding. (**A**) Model of Orf9b homodimer illustrating the Δ91–97 truncation (cyan). Residues 91–97 form 7 hydrogen bonds (7 mediated by backbone backbone interactions) between chains A and B (black dashes) of the homodimer and represent ⅔ of the homodimer binding interface. (**B**) Berkeley Madonna model (solid lines) overlaid experimental data points (circles) for the FP competition kinetic assay using Orf9b Δ91–97. (**C**) AlphaFold model of Orf9b homodimer with a 4xSGG linker (green) fusing the C-terminus of chain A (gray) to the N-terminus of chain B (magenta). (**D**) Fitted model (solid lines) overlaid with experimental data points (circles) for the FP competition kinetic assay using Orf9b 4xSGG fusion construct. (**E**) Model parameters used for solving ODE in Berkeley Madonna for both Δ91–97 and 4xSGG constructs. (**F**) Co-immunoprecipitation between Flag-Tom70 and Strep-Orf9b mutants from HEK293T cells and immunoprecipitation was performed using anti-flag magnetic beads. Representative western blots of whole-cell lysates (WCLs) and eluates after IP are shown. Actin was used as a loading control in WCLs. (**G**) Fold induction of ISRE-reporter activated by 3p-hpRNA upon expression of empty vector, Orf9b WT, Orf9b fused, Orf9b, truncated, and Orf9b S50/53E in HEK293T cells. Fold induction was calculated relative to unstimulated cells.

The online version of this article includes the following source data and figure supplement(s) for figure 5:

**Source data 1.** Original files for western blot analysis in *Figure 5F*.

**Source data 2.** PDF file containing original files for western blot analysis in *Figure 5F* with labeled bands of interest and treatments.

**Figure supplement 1.** Mutations in Orf9b variants of concern retain homodimer behavior but do not alter Tom70 binding.

**Figure supplement 1—source data 1.** Raw SDS-PAGE of Orf9b fusion homodimer from SEC elution.

**Figure supplement 1—source data 2.** PDF of SDS-PAGE for Orf9b fusion homodimer from SEC elution with molecular weights and relevant band highlighted.

with Tom70, which lends further support to our conclusion that the truncated Orf9b is acting like an 'obligate' monomer.

## A single chain fusion Orf9b homodimer indirectly reduces binding to Tom70

To create an 'obligate' homodimer, we wanted to fuse two copies of Orf9b together in a single polypeptide chain using a flexible linker. We hypothesized that such a fusion protein should reduce monomer population by promoting rapid reassociation. By examining structures of Orf9b homodimer, we observed that the N-terminus of each chain is approximately 28 Å from the C-terminus of the opposite chain (*Figure 5C*). We designed several constructs that fused the C-terminus of one Orf9b sequence to the N-terminus of a second Orf9b sequence through a Serine-Glycine-Glycine linker of variable repeats. We purified an Orf9b fusion with 4 repeats of the SGG linker (referred to as 'fusion') and observed a retention volume that aligned with the WT homodimer (centered at 70 mL;

*Figure 5—figure supplement 1B*). SDS-PAGE showed the expected band for the fusion product at ~25 kDa (*Figure 5—figure supplement 1B*). We attempted to refold the fusion homodimer to eliminate the co-purification of lipids; however, we were unable to recover the fusion homodimer. Therefore, we used the natively purified Orf9b fusion homodimer, which is, presumably, lipid-bound.

We performed our FP kinetic assay as previously described and observed a slight decrease in signal at the highest concentrations of fusion homodimer but little to no decrease at the lower concentrations. Our experimental results showed that the 60 µM and 30 uM conditions resulted in some binding to Tom70; however, the initial decrease in signal occurred slowly over the course of approximately 2–4 hours (*Figure 5D*). Unlike our lipid-bound and apo-Orf9b homodimer results, the fusion construct did not saturate Tom70 at the highest concentrations as judged by the elevated Tom70:Orf9b-FITC concentration at equilibrium.

Although our model explicitly describes homodimer dissociation into monomers as a requisite step for Orf9b binding to Tom70, we adapted it for the fusion experimental data. In this case, all model parameters other than the association and dissociation kinetics of the fluorescent probe and Tom70 were adjusted to achieve the best agreement with the experimental data. When applied to the fusion homodimer, the parameters describing homodimer dissociation into separate monomers could instead describe the dissociation of the two β-sheet domains away from each other in the tertiary structure but remaining physically linked through the linker region. We observe that the Orf9b monomer:Tom70 association rate is altered approximately 10-fold to $7.0 \times 10^3$ $M^{-1}s^{-1}$ while the dissociation rate remains unchanged compared to WT (*Figure 5E*). Further, the interconversion rates between α-helix and β-sheet conformations increase twofold and tenfold, which we interpret as the fusion construct driving the formation of the β-sheet conformation over the α-helical conformation. Additionally, the fused homodimer association rate (which can be viewed as a rate of tertiary complex formation) is reduced 160-fold compared to WT. Taken together, these values would suggest that when two copies of Orf9b are fused together into a single chain, the interconversion between secondary structures increases with the β-sheet conformation forming quickly at the expense of a slowed α-helical formation. Given that the α-helical conformation is what drives Orf9b binding to Tom70, this reduces the rate of Orf9b association to Tom70. At the same time, the increase in secondary structure interconversion comes at the cost of reducing the tertiary complex formation. Interestingly, the homodimer dissociation rate (or the tertiary unfolding rate) is also approximately 1000-fold slower than even the WT lipid-bound rate. Taken together, these parameters may suggest a model for the homodimer fusion where upon the unfolding of the tertiary homodimer structure, the fusion construct readily adopts β-sheet conformations that do not lead to the correct tertiary fold. As a result, the α-helical fold is compromised and slower to form, which in turn slows binding to Tom70.

The fusion construct is particularly useful as a comparison to Orf9b S53E mutants for impacting Tom70 binding directly or indirectly because the S53E mutation mimics the phosphorylation that directly blocks binding to Tom70 (*Thorne et al., 2022*). Collectively, the truncated, fused, and S53E full-length Orf9b constructs position us to ask about the consequences of perturbing different steps of these coupled equilibria on cellular Tom70 interactions and signaling.

## Co-immunoprecipitation and IFN assays show continued inhibition for both truncated and fused constructs

To connect our in vitro observations to cellular signaling, we tested the truncated, fused, and phosphomimetic constructs for Tom70 binding and IFN suppression. We co-transfected flag-tagged Tom70 and strep-tagged Orf9b in HEK293T cells and performed flag-pull downs (*Figure 5F*). While both WT and truncated Orf9b co-immunoprecipitated with Tom70, Orf9b S50/53E showed no interaction, as expected (*Figure 5F*). Surprisingly, we observed that the Orf9b fusion also interacted with Tom70. This suggests that even though only a small amount of the Orf9b fusion dimer showed binding to Tom70 at the highest concentrations, this may be sufficient to drive interactions in a cellular context. To test whether these interactions were sufficient for IFN signaling suppression, we turned to an ISRE reporter cell line. We transfected cells with the same Orf9b plasmids and stimulated the cells with 3p-hpRNA, which is a RIG-I agonist and activates the RIG-I mediated type 1 interferon response (*Hornung et al., 2006*; *Rehwinkel et al., 2010*). For both WT and truncated Orf9b, we observed a decrease in the fold induction of IFN (*Figure 5G*). Again, as expected, the S50/53E construct showed comparable levels

of IFN induction to the empty vector control. Surprisingly, the fusion construct showed a decrease in IFN response that was comparable to both WT and truncated Orf9b (*Figure 5G*).

We speculate that there may be two distinct possibilities for these unexpected results. First, from our in vitro kinetic assay, we can see that although the fusion construct does not saturate Tom70 like we see with the WT experiments, we still observe some binding at the highest concentrations used. Unlike the S50/53E mutant, which directly alters the microscopic rate constant between Orf9b and Tom70, our model of the fusion construct suggests that the microscopic affinity of Orf9b for Tom70 is also manipulated by rate constants that describe the interconversion between both secondary structures and tertiary structures but does not fully inhibit binding. Therefore, it is possible that even a residual interaction between Orf9b and Tom70 may be sufficient to drive a suppression of IFN without necessarily saturating Tom70.

Second, our results with the lipid-bound homodimer have shown that lipid binding alters the observed rate of Orf9b binding to Tom70 without altering the microscopic rate constants, whereas apo-homodimer produces fast binding kinetics and rapidly reaches equilibrium with Tom70. While our in vitro kinetic results working with purified protein allow us to study how full occupancy of the homodimer by lipids biases the equilibrium, we cannot control for this in cells. One possibility that emerges is that while lipid binding to the Orf9b homodimer does occur in cells (as supported by our mass spectrometry results), not all copies of the homodimer are bound to lipids. This would leave free copies of either monomeric Orf9b or apo-Orf9b homodimer available to bind to Tom70 and result in suppressed IFN signaling. Further, our SPR results with the refolded Orf9b and Tom70 (*Figure 4C*) suggest that the interaction between Orf9b and Tom70 is fairly tight, which may result in Orf9b becoming kinetically trapped to Tom70 once it binds.

## Mutations in VOC modulate Orf9b homodimer stability

We next applied our model to study Orf9b mutations present in variants of concern (VOC). While the Delta variant possesses a T60A mutation, which is within the region of Orf9b that folds into the helix to bind Tom70, the Lambda and Omicron variants have mutations outside the Tom70 binding region. They share a P10S mutation, with Omicron also having a truncation of Δ27–29 within a disordered loop of the homodimer (*Figure 6A*). We used our assays to determine if any of these mutations altered the homodimer stability or the affinity of Orf9b for Tom70. We purified all three variants and subjected them to denaturation and refolding as previously described. All three refolded variants purified as homodimers based on their retention volumes by SEC relative to WT (*Figure 6—figure supplement 1A*). We then performed the same kinetic FP assay as previously described.

We hypothesized that the most noticeable effect these mutations would have is on the dissociation rate of the homodimer into monomers. With that in mind, to minimize overfitting, we kept all model parameters that we used for the apo-WT homodimer the same and only modulated the homodimer $k_{off}$ rate to estimate the effect these mutations have on Orf9b homodimer stability. From our model, we observed that Omicron (P10S, Δ27–29) could be modeled with a homodimer dissociation rate of $2.5 \times 10^{-1}$ $s^{-1}$ which yielded a modeled $K_D$ of 3.1 μM which is close to the experimentally determined $K_D$ of the refolded WT dimer suggesting that the P10S, Δ27–29 mutations may have a minor effect on homodimer stability (*Figure 6B*). For Lambda and Delta variants, we observed that our experimental data required either lowering the homodimer dissociation rate or increasing it to best explain our experimental data. Lambda (P10S) exhibited the most stable behavior as judged by the modeled homodimer dissociation rate of $2.0 \times 10^{-2}$ $s^{-1}$ (modeled $K_D$ = 0.25 uM) which is approximately tenfold tighter than both WT and Omicron datasets (*Figure 6C*). The Delta variant (T60A) showed a kinetic behavior resembling a single exponential decay like the Orf9b peptide, suggesting that the homodimer was much weaker. Our first attempts at fitting the Delta variant FP data was performed assuming that Orf9b T60A was completely homodimeric to begin with which yielded poor fits. We then revised our initial model to include that starting solution as a mix of monomers and homodimers at equilibrium, which we determined algebraically using the modeled dimer dissociation constant that yielded the closest agreement to our FP data. We found that a dissociation constant of 12 μM (calculated by increasing homodimer $k_{off}$ to 0.9 $s^{-1}$) with a solution that is at equilibrium between homodimer and monomers prior to chasing with the fluorescent complex yielded the closest fit (*Figure 6B*). This result suggests that the T60A mutation may be destabilizing the Orf9b homodimer.

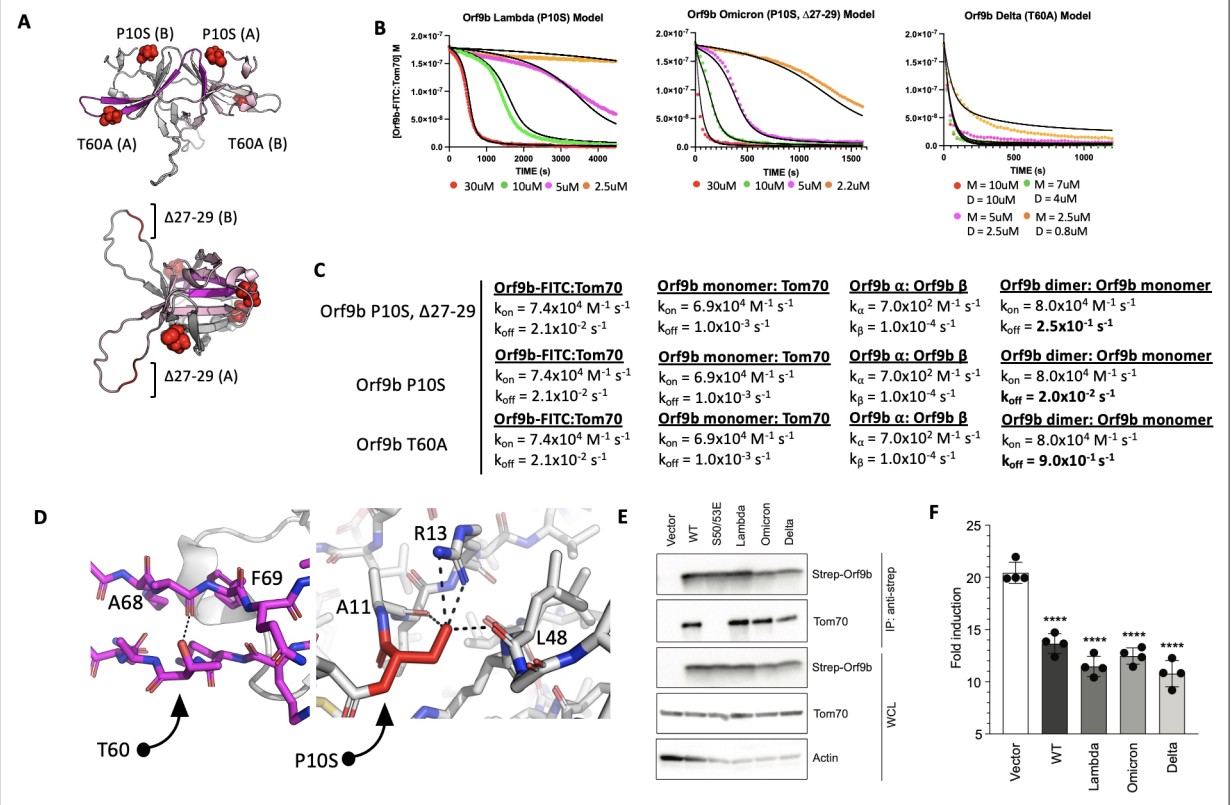

**Figure 6.** Orf9b mutations in variants of concern modulate equilibrium kinetics with Tom70. (**A**) An AlphaFold model of the full Orf9b homodimer with mutations found in variants of concern shown as red spheres. Resides that fold into the Tom70-binding α-helix are highlighted in magenta on one protomer and residues that form the dimer interface are highlighted in pink. (**B**) Berkeley Madonna model (black lines) overlaid on experimental data points (circles) for the FP competition kinetic assay using Lambda, Omicron, and Delta Orf9b variants. (**C**) Table of model parameters used to model experimental data in **B** using Berkeley Madonna. (**D**) Location of point mutations in variants of concern modeled in Pymol: (Left) T60 forms a hydrogen bond with the backbone carbonyl between A68 and F69 of the same chain that is lost in T60A. (Right) Point mutation P10S (red) introduces a serine that forms several hydrogen bonds with R13 and the backbone carbonyl of A11 and L48 within the same chain. Both mutations are mirrored on the opposite protomer. Hydrogen bonds are shown as black dashes. Mutations are modeled in Pymol using point mutation. (**E**) Co-immunoprecipitation of endogenous Tom70 with Strep-tagged Orf9b from variants of concern in HEK293T cells. (**F**) Fold induction of ISRE-reporter activated by 3p-hpRNA in HEK293T cells expressing empty vector, Orf9b WT, Orf9b from VOC (Delta, Lambda, and Omicron).

The online version of this article includes the following source data and figure supplement(s) for figure 6:

**Source data 1.** Original files for western blot analysis in **Figure 6E**.

**Source data 2.** PDF file containing original files for western blot analysis in **Figure 6E** with labeled bands of interest and treatments.

**Figure supplement 1.** Biological replicates of Orf9b variants with model fits and model parameters used.

**Figure supplement 2.** Biological replicates of all Orf9b variants and constructs performed in FP kinetic assay with model results (solid black lines) and parameters used.

Given that the T60A mutation is also present within the β-sheet to α-helix folding region that binds to Tom70 (**Figure 6D**), we sought to confirm if this mutation also affected the microscopic affinity of Orf9b for Tom70. We generated a T60A peptide based on our previously described peptide and performed SPR. We hypothesized that if the T60A mutation resulted in a weaker affinity for Tom70, then that would be reflected in the SPR compared to the WT peptide. We repeated the SPR experiment as previously for WT (**Figure 1B**) and did not observe a decrease in the affinity between Orf9b T60A and Tom70. We interpret this to mean that the T60A mutation does not reduce the affinity between Orf9b and Tom70 (**Figure 6—figure supplement 1B**), which was concordant with our model parameters where we left the Orf9b monomer:Tom70 dissociation rate unchanged (**Figure 6C**). Inspecting the location of these mutations in the Orf9b homodimer led to several structural hypotheses for the observed and modeled kinetics. For Delta, the T60A mutation is present within the β-strand that is responsible for refolding into the α-helix and binding to Tom70 (**Figure 6D**). T60 makes a hydrogen

bond with the backbone carbonyl between A68 and F69 of the same chain which may contribute to the local stability of the β-strand. For Lambda and Omicron, the P10S mutation results in the serine being positioned to form several hydrogen bonds between R13 and the backbone carbonyl of A11 and L48 within the same chain, which may contribute to local stability. The Omicron variant additionally has a truncation of residues 27–29 within a disordered loop. Although the modeled homodimer dissociation rate was close to WT, the modeled $K_D$ was slightly lower, which was consistent between biological replicates, suggesting that the P10S mutation is stabilizing the homodimer like the Lambda variant, but the truncation of residues 27–29 may counteract this effect by destabilizing the homodimer. We repeated this experiment with biological replicates for all variants of Orf9b described and observed consistent behavior (*Figure 6—figure supplement 2*).

## Mutations in VOC do not affect binding to Tom70 or IFN suppression in cell-based assays relative to WT

Our in vitro kinetic model results showed that while mutations to Orf9b present in VOC can alter the stability of the Orf9b homodimer as modeled by the homodimer dissociation rate, they did not alter the association rate of the Orf9b:Tom70 complex (*Figure 6C*). We returned to our co-IP experiments to validate if variants of Orf9b retain their interaction with Tom70 in a cellular context. We performed Co-IP of the three variants as previously described and observed no noticeable differences in Tom70 binding across different VOCs, confirming that these mutations do not directly alter the affinity between Orf9b and Tom70 (*Figure 6E*). These results were concordant with the predicted behavior that we modeled in our FP results; therefore, we hypothesized that Orf9b variants should have similar effects on IFN suppression compared to WT in our IFN assay. We performed the IFN assay as previously described and saw that compared to WT, all variants had a comparable decrease in the fold-induction of IFN, which was approximately 50% reduction compared to the empty vector control (*Figure 6F*). These results lead us to conclude that the mutations present in variants of concern may alter the Orf9b homodimerization kinetics, but they do not noticeably affect the interaction between Orf9b and Tom70 or Orf9b-mediated suppression of type 1 interferon response.

## Discussion

The SARS-CoV and 2 accessory protein Orf9b is an antagonist of the type 1 interferon response through its interactions with the mitochondrial receptor Tom70 (*Gordon et al., 2020*; *Thorne et al., 2022*; *Jiang et al., 2020*). When bound to Tom70, Orf9b adopts an amphipathic α-helical fold; however, it is also capable of binding to a second copy of Orf9b to form a β-sheet-rich homodimer. In the homodimer conformation, Orf9b can bind to lipids such as 1-palmitolyl-sn-glycerol, a precursor to palmitate which is a metabolite that is known to be involved in innate immune activation (*Meier et al., 2006*; *Jin et al., 2023*). These two different Orf9b conformations suggest that an equilibrium exists between homodimeric and monomeric conformations, with only the monomeric conformation capable of binding to Tom70. How this equilibrium is established and regulated may be of interest for both developing therapeutics as well as learning how the virus hijacks the host cell processes during infection.

We observed that the Orf9b equilibrium is heavily biased towards the Orf9b homodimer upon binding of lipids, which Orf9b co-purifies with from both mammalian cells and *E. coli*. By determining the microscopic rate constants of the Orf9b-Tom70 equilibrium using SPR, we were able to fit ODEs describing the coupled equilibrium between Orf9b conformations and Tom70 to quantify the contribution of lipid binding to the Orf9b-Tom70 equilibrium. Specifically, we have shown that lipid binding to the Orf9b homodimer slows the observed rate of binding to Tom70 without changing the microscopic rate constant of Orf9b binding to Tom70. Rather, it does so by reducing the rate of Orf9b homodimer dissociation approximately 100-fold relative to apo-homodimer. When analyzing our structural data of the refolded Orf9b homodimer, we see that lipid-binding does not drive Orf9b homodimerization as we are able to recover the homodimer conformation observed in lipid-bound structures without the lipid present. The apo-Orf9b homodimer retains the same conformation as the lipid-bound structure, but we also observed an increased level of disorder as judged by the high B-factors and lower resolution of our structure when compared to published structures with lipids bound. We speculate that this lends further evidence to our

argument that the lipid is not responsible for homodimerization but tightly stabilizes the homodimer conformation when compared to existing structures of the lipid-bound SARS CoV 2 Orf9b homodimer.

We also found that the Orf9b equilibrium could be driven towards either the monomer or homodimer by truncating seven amino acids or by fusing two copies of Orf9b together, producing behaviors that were either entirely monomeric or largely homodimeric in the context of Tom70-binding. Unlike with lipid-bound WT homodimer, the lipid-bound homodimer fusion construct did not saturate Tom70 in our kinetic assay, which suggested that the fusion construct was acting as an obligate homodimer. However, when we tested the fusion homodimer construct in both Co-IP's and IFN signaling assays, we observed that the homodimer fusion construct continued to bind to Tom70 leading to a suppressed IFN induction like WT. We had anticipated that if the Orf9b was an obligate homodimer, we would observe results resembling the phosphomimetic S53E construct, which does not bind Tom70 (*Thorne et al., 2022*) but does retain the ability to homodimerize. This result was unexpected as we had anticipated the obligate homodimer results to resemble the phosphomimetic. We hypothesize that this may be explained by two possible factors. First, we can't exclude the introduction of an increased avidity between Orf9b and Tom70 when using the fused homodimer. Although our modeled decrease in the association rate of Orf9b:Tom70 (which increases the $K_D$ of the complex) suggests that fusing two copies of Orf9b decreases the affinity to Tom70, one copy of the fusion construct could also be capable of either binding to two copies of Tom70, or, one copy of the fusion could undergo rapid rebinding to Tom70. These effects would lead to a much tighter interaction in cellular assays than we modeled in vitro. A second possible explanation is that our assumptions about high lipid binding are not valid for cell-based assays.

As we have shown with both WT and fusion constructs, recombinantly expressed and purified Orf9b is lipid-bound, and this can stabilize the homodimer to slow or inhibit the binding to Tom70. For the Orf9b fusion construct, we attempted to isolate the lipid-free species through protein refolding as previously described to compare the effect of lipid-binding on the homodimer fusion (similar to our WT experiments); however, we could not recover the stably folded homodimer. We hypothesize that the discrepancy between our kinetic results and Co-IP/IFN results could be due to subsaturation of the Orf9b fusion homodimers by lipids in cell-based assays. While we have shown that lipid-binding occurs in recombinant expression systems, it is possible that in our cell-based signaling assays, lipid-binding only affects a minor population of Orf9b. Given that we were unable to isolate the apo-fusion homodimer, we could not directly compare whether there are differences in fusion homodimer stability in the presence or absence of lipid-binding. Therefore, it is possible that the apo-fusion homodimer undergoes unfolding and refolding into alpha helices that lead to Tom70 binding similar to the WT construct. A possible future experimental direction could be attempting to pre-incubate cells with excess palmitate prior to transfection with Tom70 and Orf9b to increase the concentration of the lipid-bound Orf9b homodimer and observe its effect on Tom70 pull-down and IFN signaling.

We have also shown how mutations in VOC can perturb different steps in the Orf9b-Tom70 equilibrium. Orf9b is encoded through an alternative open reading frame within the N gene of SARS CoV 1 and 2, which results in coding mutations to N being picked up by the coding region of Orf9b. These mutations include Omicron (N: P13L; Orf9b: P10S), Delta (N: D63G; Of9b: T60A), and Lambda (N:Δ30–32, P13L; Orf9b:Δ27–29, P10S) (*Thorne et al., 2022*). Our model and FP results suggest that in the absence of lipid-binding, Orf9b VOCs still achieve rapid equilibration with Tom70 with no change in overall affinity, which is concordant with our cell-based assay results. We have shown that mutations present in Lambda and Delta can be stabilizing or destabilizing to the Orf9b homodimer and shift the time it takes to reach equilibrium with Tom70 in our kinetic assay. Omicron possesses both a truncation of residues 27–29 as well as the P10S point mutation that is present in Lambda, but shows a kinetic behavior that more closely resembles WT homodimer than Lambda homodimer. These mutations may cancel their effects out where the proposed stabilizing effect of the P10S mutation observed in Lambda is countered by a destabilizing effect due to the truncation of residues 27–29. Although none of the mutations in Delta, Lambda, and Omicron variants are found at the homodimer binding interface, we speculate that their effect on Orf9b homodimer stability is exercised through local stabilizing or destabilizing forces on the overall homodimer conformation. We cannot exclude the possibility that these mutations may primarily be tuned to affect nucleocapsid function rather than Orf9b. At this time, the effect of these specific mutations on nucleocapsid function has not

been assayed; therefore, we cannot conclude whether they produce a noticeable effect on viral RNA packaging.

When considering our results together, we propose the following expanded model for Orf9b-mediated suppression of IFN through Tom70 (*Figure 7*). During viral infection, Orf9b can either bind to Tom70 as a monomer or homodimerize with a second copy of Orf9b. In the absence of lipid-binding, Orf9b homodimers rapidly equilibrate into monomers which can subsequently bind to Tom70, creating two pathways for freshly translated Orf9b to engage Tom70: (1) Orf9b monomers immediately bind to Tom70 after translation or (2) they homodimerize with a second copy of Orf9b, which then can undergo dissociation into monomers and bind Tom70. The binding kinetics of Orf9b:Tom70 and Orf9b:Orf9b (as determined by SPR) suggest that copies of Orf9b bound to Tom70 remain tightly associated, whereas the homodimers can quickly collapse into monomers in the absence of lipid-binding. Upon lipid binding, the Orf9b homodimer is stabilized; however, they can still undergo dissociation into monomers but on a much longer time scale. Therefore, if lipid binding occurs either at a later stage of infection or is sub-saturating the Orf9b homodimer population, it may not noticeably affect IFN signaling as the copies of Orf9b bound to Tom70 remain tightly associated. This is supported in two ways: First, our VOC data shows that modulating the homodimer stability does not drive a noticeable increase or decrease in the amount of Orf9b that binds to Tom70 nor subsequent IFN signaling levels relative to WT. Second, our truncated Orf9b (obligate monomer) does not form a stable homodimer but still achieves comparable levels of IFN suppression as WT, which can homodimerize. Given that the nature of lipid-binding to the Orf9b homodimer remains unknown, we cannot precisely place when lipid-binding becomes relevant to the Orf9b-Tom70 equilibrium in the context of IFN signaling. If lipid binding is a later event during infection, then it might not be fully captured in our assays which occur over the course of either minutes or hours. Further, the tight affinity between Orf9b and Tom70 may suggest that upon binding, Orf9b becomes kinetically trapped to Tom70 and does not fully revert to the homodimer conformation, resulting in the immediate suppression of IFN.

Although we have not directly tested for the role the homodimer conformation plays during infection, we have demonstrated that lipid binding to the homodimer can bias the equilibrium away from Tom70. Lipids, including palmitate, have been shown to act as both a signaling molecule as well as a post-translational modification during antiviral innate immune signaling (*Abrami et al., 2024*; *Wen et al., 2022*; *Yang et al., 2019*). As a post-translational modification (referred to as S-acylation), MAVS, a mitochondrial type 1 IFN signaling protein that associates with Tom70 (*Liu et al., 2010*; *McWhirter et al., 2005*; *Seth et al., 2005*), has been shown to be post-translationally palmitoylated, which affects its ability to localize to the mitochondrial outer membrane during viral infection and is a known target of Orf9b (*Bu et al., 2024*; *Lee et al., 2025*). When this is impaired (either by mutation or by depletion of the palmitoylation enzyme ZDHHC24), IFN activation is impaired (*Bu et al., 2024*). Therefore, future investigations should consider if the homodimer conformation of Orf9b is capable of

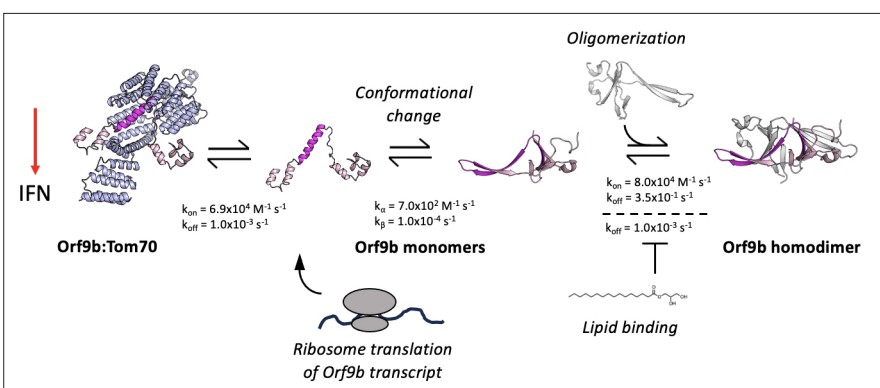

**Figure 7.** Proposed model of the Orf9b-Tom70 equilibrium. Orf9b transcripts are translated to produce monomeric Orf9b proteins that can bind to Tom70 leading to a suppression of IFN. Monomeric Orf9b can also undergo a conformational change and bind to a second copy of Orf9b to form the homodimer. The Orf9b homodimer is in equilibrium between free Orf9b monomers and Orf9b monomers bound to Tom70. Upon lipid binding, the homodimer is tightly stabilized and slowly releases monomeric Orf9b.

antagonizing other IFN signaling factors such as MAVS by binding to palmitoyl groups. Indeed, Orf9b has already been shown to be capable of binding to MAVS by Co-IP (*Han et al., 2021*); however, whether or not this occurs through the palmitoyl modification remains unknown. Collectively, our work points to the importance of this lipid-binding role of Orf9b in modulating its coupled equilibria and likely its influence on IFN signaling in the context of viral infection.

## Materials and methods

### Expression and purification of mammalian expressed WT Orf9b

WT Orf9b was transiently transfected into Expi293 cells seeded to a density of $3 \times 10^6$ cells/mL and expressed for 3 days. Harvested cells were centrifuged and resuspended in a lysis buffer of 150 mM NaCl, 50 mM Tris-HCl pH 8, 1.5 mM $MgCl_2$, 1 mM EDTA, and 0.5 mM TCEP, lysed by sonication, and then centrifuged. Supernatant was transferred to a column of Streptactin 4-flow and allowed to bind at 4 C for 1–2 hr. The column was washed with a buffer of 150 mM NaCl, 50 mM Tris-HCl pH 8, and 0.5 mM TCEP followed by elution of the strep-tagged protein with BXT buffer. Eluted strep-tagged Orf9b was ion exchanged on a CaptoQ column equilibrated with 20 mM Tris-HCl pH 8, 50 mM NaCl, and 0.5 mM TCEP with a salt gradient up to 1 mM NaCl. Ion exchanged Orf9b was then sized on a Superdex 75 10/300 GL column equilibrated with 150 mM Tris pH 8, 50 mM NaCl, and 0.5 mM TCEP.

### Purification of native WT Orf9b

The WT Orf9b gene was cloned into a pET-29a backbone with an N-terminal 6xHis tag and TEV protease cleavage site. BL21 *E. coli* cells from NEB were transformed and grown in a starter culture of Luria Broth at 37 °C overnight. 1 L cultures were inoculated with 10 mL of the starter culture and grown in Terrific Broth media supplemented with Kanamycin (100 μg/mL) at 37 C until an optical density of 0.6–0.8 was reached. Cultures were chilled at 4 °C for 15 min and then induced with 1 mM IPTG and grown at 16 °C overnight. Frozen cell pellets were resuspended in a lysis buffer of 300 mM NaCl, 50 mM Tris-HCl pH 7.5, and 0.5 mM TCEP and a cOMPLETE EDTA-free protease inhibitor tablet added per liter followed by sonication. Lysate was clarified by centrifugation and loaded onto a 5 mL Ni-NTA column pre-equilibrated with the lysis buffer supplemented with 10 mM Imidazole. The tagged protein was eluted off the column with a buffer of 300 mM NaCl, 50 mM Tris-HCl pH 7.5, 300 mM imidazole, and 0.5 mM TCEP. The 6xHis tag was cleaved by TEV protease added in a ratio of 1:20 by mass and dialyzed into a buffer of 150 mM NaCl, 50 mM Tris-HCl pH 7.5, 5% glycerol, and 0.5 mM TCEP overnight. The cleaved protein was separated by rerunning the cleaving reaction over a Ni-NTA column and collecting the flow through. The cleaved protein was then concentrated with a 3 kDa Amicon centrifugal filter and purified by size exclusion chromatography using a Superdex 75 16/600 column equilibrated with 100 mM NaCl, 20 mM HEPES pH 7.5, and 5% glycerol. Eluted fractions were pooled, filtered, and flash frozen in liquid nitrogen and stored at –80 °C.

### Purification of refolded Orf9b

Orf9b (WT, Delta, Lambda, Omicron) was cloned into a pET-29a backbone with an N-terminal 6xHis tag and TEV protease cleavage site. BL21 *E. coli* cells from NEB were transformed and grown in a starter culture of Luria Broth at 37 °C overnight. 1 L cultures were inoculated with 10 mL of the starter culture and grown in Terrific Broth media supplemented with Kanamycin (100 μg/mL) at 37 °C until an optical density of 0.6–0.8 was reached. Cultures were then induced with 1 mM IPTG and grown at 16 °C overnight. Frozen cell pellets were resuspended in a lysis buffer of 300 mM NaCl, 50 mM Tris-HCl pH 7.5, and a cOMPLETE EDTA-free protease inhibitor tablet added per liter followed by sonication. Lysate was clarified by centrifugation and loaded onto a 5 mL HisTrap HP Ni-NTA column pre-equilibrated with the lysis buffer supplemented with 10 mM Imidazole. 10 column volumes of buffer (6 M Guanidine HCl, 300 mM NaCl, 50 mM Tris-HCl pH 7.5, and 0.1% Triton X-1000, 0.5 mm TCEP) was flowed and the tagged protein was eluted with 6 M Guanidine HCl, 300 mM NaCl, 50 mM Tris-HCl pH 7.5, 300 mM imidazole, and 0.5 mM TCEP. The denatured tagged Orf9b was diluted to less than 65 μg/mL in a refolding buffer composed of 550 mM Guanidine HCl, 55 mM Tris, 21 mM NaCl, 0.88 mM KCl at pH 8.2 overnight at 4 °C and allowed to refold. The refolded 6xHis tag was cleaved by TEV protease, added in a ratio of 1:20 by mass and dialyzed into a buffer of 150 mM NaCl, 50 mM Tris-HCl pH 7.5, and 0.5 mM TCEP overnight. The cleaved protein was separated by rerunning

the cleaving reaction over a Ni-NTA column and collecting the flow through. The cleaved protein was then concentrated with a 3 kDa Amicon centrifugal filter and purified by size exclusion chromatography using a Superdex 75 16/600 column equilibrated with 100 mM NaCl and 20 mM HEPES pH 7.5 and 5% glycerol.

## Purification of hTom70

hTom70 (109-600) lacking the transmembrane domain was cloned into a pET-29b(+) backbone with an N-terminal 6xHis tag and SUMO tag. Transformation and growing conditions were the same as WT Orf9b other than the media being supplemented with Carbenicillin (100 μg/mL). Ni-NTA column purification conditions and buffers were also the same as native Orf9b except for the cleavage of the 6x-His and SUMO tag which was performed with Ulp1 protease at a ratio of 1:20 by mass. All other conditions were the same except for size exclusion chromatography which was performed with a Superdex 200 16/600 column.

## Generating biotinylated Tom70

The Tom70 (109-600) plasmid was used for generating biotinylated Tom70 for SPR experiments. An AVI tag was introduced to the N-terminus using a Q5 Site-Directed Mutagenesis kit (NEB: E0544S) and confirmed by sequencing. BL21 cells were cotransfected with the AVI-tagged plasmid and BirA biotin ligase plasmid in equal quantities by mass. Tom70 AVI-tagged plasmid was carbenicillin resistant, and BirA was chloramphenicol resistant for double selection. Starter cultures were grown ON at 37C in LB media. 1 L cultures were inoculated with 10 mL of the starter culture and grown in Terrific Broth media supplemented with both chloramphenicol and carbenicillin (100 μg/mL) and grown at 37 °C until an optical density of 0.6–0.8 was reached. 1 L cultures were then induced with 1 mM IPTG and 100 μM biotin and grown overnight at 16 °C. Purification proceeded as previously described with the modification of not including a TEV cleavage reaction.

## Generation and purification of Orf9b mutants

Orf9b mutants (Δ91–97 and 4xSGG) were generated by mutagenesis using the WT Orf9b plasmid. Mutations were introduced by primers using a Q5 Site-Directed Mutagenesis kit (NEB: E0544S). Mutants were transformed into NEB 5-alpha cells and colonies grown in 10 mL of LB overnight at 37 °C. Mutants were subjected to miniprep and submitted for sequencing to confirm the mutation's presence. Growth and purification conditions of all constructs were the same as WT Orf9b.

## FP kinetic experiments

Orf9b-FITC was synthesized by Biomatik comprising residues 44–70 of Orf9b with a C-terminal fluorescein fluorophore. Orf9b-FITC was resuspended from lyophilized powder in a 3:1 ratio of 100 mM NaCl, 20 mM HEPES pH 7.5 and DMSO and filtered. 2 μM of Orf9b-FITC was mixed with 25 μM Tom70 in a buffer of 100 mM NaCl, 20 mM HEPES pH 7.5, 0.5 mM TCEP to generate the 10x fluorescent Orf9b-FITC:Tom70 complex in black eppendorf tubes and equilibrated for 1 hr. 90 μL of Orf9b (peptide, refolded or native) was added to black 96 well plates with a non-binding surface (Greiner: 655900). 10 μL of the 10x Orf9b-FITC:Tom70 complex was then added to initiate the kinetic experiment. A solution of 200 nM Orf9b-FITC in buffer was used to determine the minimal FP signal. We chased the pre-equilibrated Orf9b-FITC:Tom70 fluorescent complex with either unlabeled Orf9b peptides or Orf9b homodimer (WT, VOC, and fusion) and monitored the decrease in polarization until equilibrium was reached. Measurements were performed on a Tecan Spark running Magellan with a monochromator (excitation: 485 nm, emission: 535 nm). Measurements were taken in 15 s intervals between measurements with 7 s of shaking between intervals. Biological replicates of the kinetic assay were performed by repurifying all Orf9b variants presented following the denaturing and refolding.

## Refolded and lipid-bound Orf9b homodimer crystallization and refinement

TEV cleaved refolded Orf9b was concentrated to 10 mg/ml and crystallized by sitting drop diffusion with a reservoir buffer of 1.2 M Sodium Phosphate, 0.8 M Potassium Phosphate, 0.2 M Lithium Sulfate, and 0.1 M Glycine pH 10.5 at 20 °C. Crystallization drops were set up with 200 nL of Orf9b mixed with 200 nL of the mother liquor in SWISSCI 2-well plates. TEV cleaved natively folded Orf9b was

concentrated to 10 mg/ml and crystallized by sitting drop diffusion with a reservoir buffer of 10% PEG 3350, 10% PEG 1000, 10% MPD, 100 mM Imidazole pH 6.5, 100 M MES pH 6.5, and 150 mM Ethylene glycol at 20°C. Crystallization drops were set up with 200 nL of Orf9b mixed with 200 nL of the mother liquor in SWISSCI 2-well plates. Crystals appeared 1 day after plating and grew to a maximum size in 7 days. All crystals appeared 1 day after plating and grew to a maximum size in 7 days. Crystals were looped and flash-frozen without added cryogen. X-ray diffraction data was collected at the Advanced Light Source beamline 8.3.1. Diffraction data was indexed in XDS (**Kabsch, 2010**) and merged and scaled in XSCALE. Molecular replacement was performed in phenix.phaser (**Winn et al., 2011**) using PDB 6Z4U as a reference (**Echols et al., 2012**) with five macrocycles of refinement and was deposited under PDB 9MZB and 9N55. Polder maps were generated in phenix.polder (**Liebschner et al., 2017**).

## Surface plasmon resonance

For studying Orf9b peptide/Orf9b:Tom70 kinetics, biotinylated Tom70 was immobilized to a CMD500M chip (Xantec: SPSMCMD500M). 250 nM biotin-Tom70 was immobilized for 60 s at 25 µL/s to either spots B or C on a neutravidin coated surface followed by 120 s of dissociation. Spot A on the chip was used as a reference surface and had no Tom70 immobilized to it. Analytes used for studying Tom70 binding were exchanged into the SPR running buffer using Pierce 7 k MWCO Desalting Spin columns (Thermo: 89862). Blank injections of the running buffer were regularly interspersed between analyte injections. Binding kinetics were performed using a Bruker SPR-24 Pro. All SPR experiments were performed using a running buffer of 100 mM NaCl, 20 mM HEPES pH 7.5, 0.5 mM TCEP, and 0.05% Tween20. All binding sensorgrams were double reference subtracted from the reference surface and the running buffer blank injections. Sensorgram fitting to determine kinetic rates and $K_D$ was performed using Sierra Analyser 3 provided by Bruker.

## Kinetic modeling

Modeling of the FP binding data was performed in Berkeley Madonna v10.6.1 (**Marcoline et al., 2022**) using the Rosenbrock stiff ordinary differential equations (ODE) solver with a $10^{-2}$ relative tolerance and a maximum time step of 1 s. All data fitting used the Nelder-Mead simplex method and a root mean squared error function between the model solution and the experimental data points. Converting from fluorescent polarization values to concentration of Orf9b-FITC:Tom70 complex was accomplished using the following equation:

$$\left( \frac{Signal(t) - Background\,Signal}{Max\,FP\,signal} \right) * Conc \cdot Fluorescent\,Complex = Conc \cdot Fluorescent\,Complex\,(t)$$

where max FP signal at time *t* is the FP signal produced in the absence of unlabeled competitor and the concentration of the fluorescent complex is the maximum concentration achieved based on the concentration of Orf9b-FITC and Tom70 and the dissociation constant for the complex.

The model was initially populated with experimental values taken directly from surface plasmon resonance experiments. We then performed optimizations fitting the kinetic curves for the lowest concentrations (individually) and with additional minor adjustments to derive the best agreement with the experimental data. We used these model parameters to show the correct behavior as the concentration of Orf9b homodimer changed between conditions. For scenarios involving an initial equilibrium between monomer and dimer populations prior to the start of the experiment (Orf9b T60A variant), we solved for the steady state equilibrium concentrations algebraically and then estimated their initial starting concentrations for each condition in the ODE solver.

For sensitivity analysis to the peptide model in *Figure 1D*, we inspected each model parameter by applying a +/- 10% change to the reported values to observe their effect on the models' performance. Only the rate of Orf9b-FITC dissociation showed sensitivity to change producing the most notable differences in model performance relative to the experimental data. For the homodimer model outlined in *Figure 4Ai* and Aii, the model was most sensitive to the rate of β to α-monomer conversion and the rate of homodimer association and dissociation. For the apo-homodimer dataset, we investigated both parameters for each concentration series and observed that both parameters were sensitive to change at the lowest concentration of 5 µM but became insensitive to change at the highest concentration of 50 µM suggesting that the model parameter describing the rate of β to α-monomer conversion is not essential at the highest concentrations but remains essential at the

lowest concentrations. Further, we repeated this analysis for the lipid-bound dataset across all concentration series, which revealed that the model was sensitive to perturbation of both parameters at all concentrations. These three parameters show similar sensitivity because they exist as sequential steps in the rate-limiting part of the reaction and are sensitive to the perturbations we introduce.

## Co-immunoprecipitation assay

HEK293T cells were transfected with the indicated mammalian expression plasmids using Lipofectamine 2000 (Invitrogen). After 24 hr, the cells were harvested and lysed in NP-40 lysis buffer (0.5% Nonidet P-40 Substitute (NP-40; Fluka Analytical), 50 mM Tris-HCl (pH 7.4 at 4°C), 150 mM NaCl, and 1 mM EDTA) supplemented with cOmplete Mini EDTA-free protease inhibitor and PhosSTOP phosphatase inhibitor cocktails (Roche). The clarified lysates were incubated with anti-DYKDDDDK magnetic agarose (Pierce) or Strep-Tactin Sepharose beads (IBA) for 2 hr or overnight (for endogenous pull-downs) at 4°C, followed by five washes with NP-40 lysis buffer. Bound protein complexes were eluted in SDS loading buffer or using 3X flag peptide (Sigma) and analyzed by western blotting using the indicated antibodies.

Antibodies used: rabbit anti–Strep-tag II (Abcam, ab232586); rabbit anti-β-actin (Cell Signaling Technology, 4967); monoclonal mouse anti-Flag M2 (Sigma Aldrich, F1804); monoclonal mouse anti-Tom70 (Santa Cruz, sc-390545); and polyclonal rabbit anti-Flag (Sigma Aldrich, F7425).

## IFN-signaling assay

Each viral protein was overexpressed in HEK293T cells containing a Lucia reporter under the control of the IFN-β/ISG56 promoter, alongside a vector control. For viral protein expression, cells were transfected with 250 ng of either an empty vector or a vector encoding Orf9b WT or mutants using Lipofectamine 2000 (Invitrogen). Twenty-four hours after plasmid transfection, cells were stimulated with 3p-hpRNA (InvivoGen), a RIG-I agonist. After an additional 24 hours, 20 µL of cell media (supernatant) from each well was transferred into a 96-well white (opaque) plate and mixed with 50 µL of QUANTI-Luc 4 Lucia/Gaussia 'Glow' solution (InvivoGen). Lucia activity was measured by reading luminescence on a SpectraMax iD3 system, and fold activation was calculated relative to unstimulated cells.

## Native mass spectrometry

For native mass spectrometry analysis, protein samples were desalted into 200 mM ammonium acetate using a 7 kDa MWCO spin centrifugal column (Thermo Fisher Scientific) to eliminate non-volatile buffer components. The desalted protein was diluted to a final concentration of 0.5 µM in ammonium acetate and loaded into a metal-coated borosilicate nanoelectrospray emitter (Thermo Fisher Scientific) and infused into a Q Exactive Extended Mass Range (EMR) Orbitrap mass spectrometer (Thermo Fisher Scientific) in negative polarity mode. The instrument was operated with the following parameters: capillary voltage 0.9–1.5 kV, capillary temperature 200°C, and resolving power set to 17,500 (at m/z 200). Ion optics were tuned to preserve macromolecular complexes using a source DC offset of –15 V, injection flatapole DC of –10 V, inter-flatapole lens DC of –5 V, bent flatapole DC of –5 V, and transfer multipole DC of 0 V. Additional settings included 5 microscans, an automatic gain control (AGC) target of $3e^6$, S-lens RF level of 200, and trapping gas pressure of 1.0 (arbitrary units) to minimize collisional activation. Raw mass spectra were acquired across a m/z range of 900–7000 and deconvoluted using UniDec software (v5.0.2; *Marty et al., 2015*).

## Mass spectrometry

Lipid isolation was performed using a protocol derived from the Folch method (*Lees et al., 1957*). Purified protein (10 µL) was combined with a 2:1 (v/v) methanol-chloroform mixture (500 µL) and vortexed thoroughly to ensure homogeneity. The mixture was incubated at ambient temperature for 10 min to solubilize the lipids. Phase separation was induced by adding ultrapure water to adjust the solvent ratio to 1:1:0.9 (methanol:chloroform:water), followed by centrifugation at 2000 × *g* for 10 min. The lower chloroform-rich phase, enriched in lipids, was isolated, evaporated to dryness, and redissolved in 500 µL of 50:50 (v/v) acetonitrile:water supplemented with 0.1% formic acid. For mass spectrometric analysis, the reconstituted sample was directly infused into a Thermo Fisher Scientific Q Exactive Plus mass spectrometer via a syringe pump at 3 µL/min flow rate. Electrospray ionization was employed with a spray voltage of 2 kV and a capillary temperature of 320°C in the positive

polarity mode. High-resolution full-scan spectra were acquired at 140,000 resolution (m/z 200) across a range of m/z 150–1,000. Targeted lipid ions were subjected to tandem MS (MS/MS) fragmentation for structural elucidation. Data was processed using Freestyle 1.8 SP2 software. Lipid identification was performed by matching acquired spectra to theoretical isotopic distributions and fragmentation patterns using the LIPID MAPS (*Conroy et al., 2024*), Human Metabolome Database (HMDB) (*Wishart et al., 2022*), and PubChem databases (*Kim et al., 2025*).

## Acknowledgements

We acknowledge helpful comments from Jason Gestwicki and John Gross. This work was funded by NIH U19AI171110 and GM145238 (to JSF) and R01GM137109 (to MG). Native mass spectrometric experiments were conducted on the Thermo Scientific Exactive Plus EMR, funded by the UCSF Research Resource Program and NIH P41GM103481, at the Mass Spectrometry Resource at UCSF (A.L. Burlingame, Director), supported by the Dr. Miriam and Sheldon G Adelson Medical Research Foundation (AMRF) and the NIH-NIGMS. The diffraction data of structures reported in this work was collected at beamline 8.3.1. of the Advanced Light Source (ALS). The ALS, a U.S. DOE Office of Science User Facility under contract no. DE-AC02-05CH11231, is supported in part by the ALS-ENABLE program funded by the NIH, National Institute of General Medical Sciences, grant P30GM124169.

## Additional information

### Competing interests

Nevan J Krogan: The Krogan Laboratory has received research support from Vir Biotechnology, F. Hoffmann-La Roche, and Rezo Therapeutics; NK has a financially compensated consulting agreement with Maze Therapeutics; NK is the President and is on the Board of Directors of Rezo Therapeutics, and is a shareholder in Tenaya Therapeutics, Maze Therapeutics, Rezo Therapeutics, GEn1E Lifesciences, and Interline Therapeutics. Michael Grabe: Is a developer for Berkeley Madonna. James S Fraser: Is a consultant to, shareholder of, and receives sponsored research support from Relay Therapeutics. The other authors declare that no competing interests exist.

### Funding

| Funder | Grant reference number | Author |
|---|---|---|
| National Institute of Allergy and Infectious Diseases | U19AI171110 | Nevan J Krogan |
| National Institute of General Medical Sciences | GM145238 | James S Fraser |
| National Institute of General Medical Sciences | R01GM137109 | Michael Grabe |

The funders had no role in study design, data collection and interpretation, or the decision to submit the work for publication.

### Author contributions

CJ San Felipe, Conceptualization, Data curation, Formal analysis, Validation, Investigation, Visualization, Methodology, Writing – original draft, Writing – review and editing; Jyoti Batra, Monita Muralidharan, Shivali Malpotra, Durga Anand, Rachel Bauer, Data curation, Formal analysis, Validation, Investigation, Visualization, Methodology, Writing – review and editing; Kliment A Verba, Data curation, Formal analysis, Supervision, Validation, Investigation, Visualization, Methodology, Writing – review and editing; Danielle L Swaney, Data curation, Formal analysis, Supervision, Validation, Investigation, Visualization, Methodology, Writing – original draft, Writing – review and editing; Nevan J Krogan, Supervision, Funding acquisition, Writing – review and editing; Michael Grabe, Conceptualization, Resources, Data curation, Software, Formal analysis, Supervision, Funding acquisition, Validation, Investigation, Visualization, Methodology, Writing – original draft, Project administration, Writing – review and editing; James S Fraser, Conceptualization, Resources, Data curation, Formal analysis,

Supervision, Funding acquisition, Validation, Investigation, Visualization, Methodology, Writing – original draft, Project administration, Writing – review and editing

### Author ORCIDs
CJ San Felipe (iD) https://orcid.org/0000-0002-2695-5951
Jyoti Batra (iD) https://orcid.org/0000-0002-2335-0607
Kliment A Verba (iD) https://orcid.org/0000-0002-2238-8590
Danielle L Swaney (iD) https://orcid.org/0000-0001-6119-6084
Nevan J Krogan (iD) https://orcid.org/0000-0003-4902-337X
Michael Grabe (iD) https://orcid.org/0000-0003-3509-5997
James S Fraser (iD) https://orcid.org/0000-0002-5080-2859

Reviewer #1 (Public review): https://doi.org/10.7554/eLife.106484.3.sa1
Reviewer #2 (Public review): https://doi.org/10.7554/eLife.106484.3.sa2
Author response https://doi.org/10.7554/eLife.106484.3.sa3

## Additional files

### Supplementary files
Supplementary file 1. Crystallograpy table and statistics.

MDAR checklist

### Data availability
PDB depositions: 9MZB, 9N55.

The following datasets were generated:

| Author(s) | Year | Dataset title | Dataset URL | Database and Identifier |
|---|---|---|---|---|
| San Felipe CJ, Fraser JS | 2025 | X-ray crystallographic structure of Orf9b Apo Homodimer | https://doi.org/10.2210/pdb9mzb/pdb | Worldwide Protein Data Bank, 10.2210/pdb9mzb/pdb |
| San Felipe CJ, Fraser JS | 2025 | X-ray Crystallographic Structure of Lipid-bound Orf9b Homodimer | https://doi.org/10.2210/pdb9n55/pdb | Worldwide Protein Data Bank, 10.2210/pdb9n55/pdb |

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
