## [Editor Report · eLife Assessment]

This **fundamental** study demonstrates that lipid binding can regulate the dimerization state of the SARS-CoV2 Orf9b protein. The data from biophysical and cellular experiments along with mathematical modeling are **compelling**. This paper is broadly relevant to those studying coupled equilibria across all aspects of biology.

---

## [Referee Report · Reviewer #1 (Public review)]

Summary:

Felipe and colleagues try to answer an important question in Sarbecovirus Orf9b-mediated interferon signaling suppression, given that this small viral protein adopts two distinct conformations, a dimeric β-sheet-rich fold and a helix-rich monomeric fold when bound by Tom70 protein. Two Orf9b structures determined by X-ray crystallography and Cryo-EM suggest an equilibrium between the two Orf9b conformations, and it is important to understand how this equilibrium relates to its functions. To answer these questions, the authors developed a series of ordinary differential equations (ODE) describing the Orf9b conformation equilibrium between homodimers and monomers binding to Tom70. They used SPR and a fluorescent polarization (FP) peptide displacement assay to identify parameters for the equilibrium and create a theoretical model. They then used the model to characterize the effect of lipid-binding and the effects of Orf9b mutations in homodimer stability, lipid binding, and dimer-monomer equilibrium. They used their model to further analyze dimerization, lipid binding, and Orf9b-Tom70 interactions for truncated Orf9b, Orf9b fusion mutant S53E (blocking Tom70 binding), and Orf9b from a set of Sars-CoV-2 VOCs. They evaluated the ability of different Orf9b variants for binding Tom70 using Co-IP experiments and assessed their activity in suppressing IFN signaling in cells.

Overall, this work is well designed, the results are of high quality and well-presented; the results support their conclusions.

Strengths:

(1) They developed a working biophysical model for analyzing Orf9b monomer-dimer equilibrium and Tom70 binding based on SPR and FP experiments; this is an important tool for future investigation.

(2) They prepared lipid-free Orf9b homodimer and determined its crystal structure.

(3) They designed and purified obligate Orf9b monomer, fused-dimer, etc., a very important Orf9b variant for further investigations.

(4) They identified the lipid bound by Orf9b homodimer using mass spectra data.

(5) They proposed a working model of Orf9b-Tom70 equilibrium.

Weaknesses:

(1) It is difficult to understand why the obligate Orf9b dimer has similar IFN inhibition activity as the WT protein and obligate Orf9b monomer truncations.

(2) The role of Orf9b homodimer and the role of Orf9b-bound lipid in virus infection, remains unknown.

Comments on revisions:

In the revised manuscript, the authors have addressed my concerns.

---

## [Referee Report · Reviewer #2 (Public review)]

Summary:

This study focuses on Orf9b, a SARS-COV1/2 protein that regulates innate signaling through interaction with Tom70. San Felipe et al use a combination of biophysical methods to characterize the coupling between lipid-binding, dimerization, conformational change, and protein-protein-interaction equilibria for the Orf9b-Tom70 system. Their analysis provides a detailed explanation for previous observations of Orf9b function. In a cellular context, they find other factors may also be important for the biological functioning of Orf9b.

Strengths:

San Felipe et al elegantly combine structural biology, biophysics, kinetic modelling, and cellular assays, allowing detailed analysis of the Orf9b-Tom70 system. Such complex systems involving coupled equilibria are prevalent in various aspects of biology, and a quantitative description of them, while challenging, provides a detailed understanding and prediction of biological outcomes. Using SPR to guide initial estimates of the rate constants for solution measurements is an interesting approach.

Weaknesses:

This study would benefit from a more quantitative description of uncertainties in the numerous rate constants of the models, either through a detailed presentation of the sensitivity analysis or another approach such as MCMC. Quantitative uncertainty analysis, such as MCMC is not trivial for ODEs, particularly when they involve many parameters and are to be fitted to numerous data points, as is the case for this study. The authors use sensitivity analysis as an alternative, however, the results of the sensitivity analysis are not presented in detail, and I believe the authors should consider whether there is a way to present this analysis more quantitatively. For example, could the residuals for each +/-10% parameter change for the peptide model be presented as a supplementary figure, and similarly for the more complex models? Further details of the range of rate constants tested would be useful, particularly for the ka and kB parameters.

The authors build a model that incorporates an α-helix-β-sheet conformational change, but the rate constant for the conversion to the α-helix conformation is required to be second order. Although the authors provide some rationale, I do not find this satisfactorily convincing given the large number of adjustable parameters in the model and the use of manual model fitting. The authors should discuss whether there is any precedence for second-order rate constants for conformational changes in the literature. On page 14, the authors state this rate constant "had to be non-linear in the monomer β-sheet concentration" - how many other models did the authors explore? For example, would αT↔α↔αα↔ββ (i.e., conformational change before dimer dissociation) or α↔βαT↔ββ (i.e., Tom70 binding driving dimer dissociation) be other plausible models for the conformational change that do not require assumptions of second-order rate constants for the conformational change?

Overall, this study progresses the analysis of coupled equilibria and provides insights into Orf9b function.

Comments on revisions:

The authors have done a satisfactory job addressing my concerns.

Regarding my recommendations to the authors - point 7: "Orf9b-FITC:Tom70" and "PT", representing the same species, are still both used in the equations on page 14, which is confusing for anyone who may wish to re-use the model. I appreciate this is quite a subtle point but given the importance of the model for the manuscript I feel the authors should do their due diligence to ensure it is presented as clearly as possible.

---

## [Author Response]

The following is the authors’ response to the original reviews.

**Reviewer #1 (Public review):**
Summary:Felipe and colleagues try to answer an important question in Sarbecovirus Orf9b-mediated interferon signaling suppression, given that this small viral protein adopts two distinct conformations, a dimeric β-sheet-rich fold and a helix-rich monomeric fold when bound by Tom70 protein. Two Orf9b structures determined by X-ray crystallography and Cryo-EM suggest an equilibrium between the two Orf9b conformations, and it is important to understand how this equilibrium relates to its functions. To answer these questions, the authors developed a series of ordinary differential equations (ODE) describing the Orf9b conformation equilibrium between homodimers and monomers binding to Tom70. They used SPR and a fluorescent polarization (FP) peptide displacement assay to identify parameters for the equilibrium and create a theoretical model. They then used the model to characterize the effect of lipid-binding and the effects of Orf9b mutations in homodimer stability, lipid binding, and dimer-monomer equilibrium. They used their model to further analyze dimerization, lipid binding, and Orf9b-Tom70 interactions for truncated Orf9b, Orf9b fusion mutant S53E (blocking Tom70 binding), and Orf9b from a set of Sars-CoV-2 VOCs. They evaluated the ability of different Orf9b variants for binding Tom70 using Co-IP experiments and assessed their activity in suppressing IFN signaling in cells.Overall, this work is well designed, the results are of high quality and well-presented; the results support their conclusions.

We thank reviewer #1 for their thoughtful assessment of our work and their constructive feedback.

Strengths:(1) They developed a working biophysical model for analyzing Orf9b monomer-dimer equilibrium and Tom70 binding based on SPR and FP experiments; this is an important tool for future investigation.(2) They prepared lipid-free Orf9b homodimer and determined its crystal structure.(3) They designed and purified obligate Orf9b monomer, fused-dimer, etc., a very important Orf9b variant for further investigations.(4) They identified the lipid bound by Orf9b homodimer using mass spectra data.(5) They proposed a working model of Orf9b-Tom70 equilibrium.Weaknesses:(1) It is difficult to understand why the obligate Orf9b dimer has similar IFN inhibition activity as the WT protein and obligate Orf9b monomer truncations.

We thank the reviewer for their observation and agree that the obligate homodimer IFN results were not what we expected to observe given our FP kinetic results with the purified obligate homodimer and noted our surprise in the discussion. We also note that we have two possible hypotheses for why this is the case.

In our discussion, we noted the possible introduction of an increased avidity effect with fused homodimer and have improved it as follows with additions in red:

“This result was unexpected as we had anticipated the obligate homodimer results to resemble the phosphomimetic. We hypothesize that this may be explained by two possible factors. First, we can’t exclude the introduction of an increased avidity between Orf9b and Tom70 when using the fused homodimer. Although our modeled decrease in the association rate of Orf9b:Tom70 (which increases the K_D_ of the complex) suggests that fusing two copies of Orf9b decreases the affinity to Tom70, one copy of the fusion construct could also be capable of either binding to two copies of Tom70, or, one copy of the fusion could undergo rapid rebinding to Tom70. These effects would lead to a much tighter interaction in cellular assays than we modeled in vitro. A second possible explanation is that our assumptions about high lipid binding are not valid for cell based assays.”

We also noted that a second possible explanation is due to our limitations in isolating the apo-fused homodimer to compare to the lipid-bound fused homodimer and possible differences this could have on our assays and briefly expanded upon this. Again, we improved this with additions in red:

“As we have shown with both WT and fusion constructs, recombinantly expressed and purified Orf9b is lipid-bound and this can stabilize the homodimer to slow or inhibit the binding to Tom70. For the Orf9b fusion construct, we attempted to isolate the lipid-free species through protein refolding as previously described to compare the effect of lipid-binding on the homodimer fusion (similar to our WT experiments); however, we could not recover the stably folded homodimer. We hypothesize that the discrepancy between our kinetic results and Co-IP/IFN results could be due to subsaturation of the Orf9b fusion homodimers by lipids in cell based assays. While we have shown that lipid-binding occurs in recombinant expression systems, it is possible that in our cell based signaling assays that lipid-binding only affects a minor population of Orf9b. Given that we were unable to isolate the apo-fusion homodimer, we could not directly compare whether there are differences in fusion homodimer stability in the presence or absence of lipid-binding. Therefore, it is possible that the apo-fusion homodimer undergoes unfolding and refolding into alpha helices that lead to Tom70 binding similar to the WT construct.”

(2) The role of Orf9b homodimer and the role of Orf9b-bound lipid in virus infection, remains unknown.We agree that we did not try to directly test for the role of the homodimer during infection and this remains an open area of exploration for future studies. We have included this caveat in our discussion but suggested possible experiments and future directions that could help shed light on this:

“Although we have not directly tested for the role the homodimer conformation plays during infection, we have demonstrated that lipid-binding to the homodimer can bias the equilibrium away from Tom70. Lipids including palmitate have been shown to act as both a signaling molecule as well as a post-translational modification during antiviral innate immune signaling (S Mesquita et al. 2024; Wen et al. 2022; S. Yang et al. 2019). As a post-translational modification (referred to as S-acylation), MAVS, a mitochondrial type 1 IFN signaling protein that associates with Tom70 (X.-Y. Liu et al. 2010; McWhirter, Tenoever, and Maniatis 2005; Seth et al. 2005), has been shown to be post-translationally palmitoylated which affects its ability to localize to the mitochondrial outer membrane during viral infection and is a known target of Orf9b (Bu et al. 2024; Lee et al. 2024). When this is impaired (either by mutation or by depletion of the palmitoylation enzyme ZDHHC24), IFN activation is impaired (Bu et al. 2024). Therefore, future investigations should consider if the homodimer conformation of Orf9b is capable of antagonizing other IFN signaling factors such as MAVS by binding to palmitoyl groups. Indeed, Orf9b has already been shown to be capable of binding to MAVS by Co-IP (Han et al. 2021), however, whether or not this occurs through the palmitoyl modification remains unknown.”

**Reviewer #2 (Public review):**
Summary:This study focuses on Orf9b, a SARS-COV1/2 protein that regulates innate signaling through interaction with Tom70. San Felipe et al use a combination of biophysical methods to characterize the coupling between lipid-binding, dimerization, conformational change, and protein-protein-interaction equilibria for the Orf9b-Tom70 system. Their analysis provides a detailed explanation for previous observations of Orf9b function. In a cellular context, they find other factors may also be important for the biological functioning of Orf9b.Strengths:San Felipe et al elegantly combine structural biology, biophysics, kinetic modelling, and cellular assays, allowing detailed analysis of the Orf9b-Tom70 system. Such complex systems involving coupled equilibria are prevalent in various aspects of biology, and a quantitative description of them, while challenging, provides a detailed understanding and prediction of biological outcomes. Using SPR to guide initial estimates of the rate constants for solution measurements is an interesting approach.Weaknesses:This study would benefit from a more quantitative description of uncertainties in the numerous rate constants of the models, either through a detailed presentation of the sensitivity analysis or another approach such as MCMC. Quantitative uncertainty analysis, such as MCMC is not trivial for ODEs, particularly when they involve many parameters and are to be fitted to numerous data points, as is the case for this study. The authors use sensitivity analysis as an alternative, however, the results of the sensitivity analysis are not presented in detail, and I believe the authors should consider whether there is a way to present this analysis more quantitatively. For example, could the residuals for each +/-10% parameter change for the peptide model be presented as a supplementary figure, and similarly for the more complex models? Further details of the range of rate constants tested would be useful, particularly for the ka and kB parameters.

We thank the reviewer for their constructive feedback and have generated supplemental figures providing a deeper analysis of the residuals for each model parameter adjusted +/- 10% from the reported values which we have added to our supplemental figures as Figure 1 - Supplemental 3 and Figure 4 - Supplemental 5 .

We note that there are modest improvements in residual plots where model parameters are individually lowered by 10% from their reported value when considering this single dataset, however, our choice of using the reported values was driven by finding values that were suitable for improving model behavior across multiple concentration series in different datasets. Specifically, we have also included the RMSD values for each model parameter subjected to a +/-10% change from a single concentration time course as well as the percent change in RMSD relative to the RMSD generated by our reported model parameters to illustrate this. We have also included text that makes note of the observed pattern in the residuals from Figure 4 - Supplement 5 and provided some explanations for why this may occur.

“Inspection of the residuals from the 5uM apo-Orf9b homodimer time course showed clear patterns when individual model parameters were subjected to a 10% increase or decrease from the reported values. While our proposed model qualitatively describes the concentration dependent change in kinetic behavior, the residual plots may suggest that additional binding reactions may also be occurring that are not captured by our model.”

Figure 1 - Supplemental 3. Plots of residuals from Orf9b peptide model showing effect of an increase or decrease by 10% on each model parameter. All residuals and reporting are with respect to the100uM of unlabeled Orf9b peptide condition. Blue dots: reported value. Red dots: 10% increase in reported value. Green dots: 10% decrease in reported value. Table reporting of RMSD values for model fitsafter +/-10% change to model parameter (Left column) and percent change in RMSD relative to reported model RMSD (Right column).

“As an alternative to attempting to place CIs on the parameters, we performed sensitivity analysis to determine which parameters the model was most sensitive to (see methods and Figure 1 - Supplemental 3). Additionally, we note that the model parameters were derived from the fit of only one concentration (100uM), but fit the other concentrations equally well. We observed that the model parameter that was most sensitive to change was the rate of Orf9b-FITC:Tom70 ([PT]) dissociation when subjected to a 10% increase or decrease whereas all other model parameters showed no sensitivity to change (Figure 1 - Supplemental 3).”

Figure 4 - Supplemental 5: Plot of residuals showing the effect of increasing or decreasing individual model parameters 10% compared to the reported values. All residual plots are with respect to the 5uM apo-Orf9b homodimer condition. Blue dots: reported value. Red dot: 10% increase in reported value. Green dot: 10% decrease in reported value. (Left columns) Table of RMSD values calculated from model fits showing the effect of both +/-10% change to individual model parameters. (Right columns) Percent change in RMSD values subjected to +/-10% change for individual model parameters relative to the RMSD of the reported model.

We have also included the following revised text to accompany this figure.

“Further, we repeated the sensitivity analysis described previously for the peptide model and also considered the sensitivity of model parameters by inspecting each individually (Figure 4- figure supplemental 5). We found that when examining the residuals of the lowest concentration of 5uM, the model was most sensitive to changes in three parameters: the rate of homodimer association and dissociation and the conversion from β to α-monomers.”

“Therefore, under low concentrations of Orf9b homodimer, binding to Tom70 is limited by the rate of homodimer association and dissociation as well as the conversion of Orf9b monomers to the α-helical conformation.”

We have also included a supplemental figure showing how changes in the model parameters ka and kB affect the models behavior to help illustrate the range of values tested as Figure 4 - Supplemental 4.

Figure 4 - Supplemental 4: Plots of model behavior showing the effect of changes to alpha-beta and beta-alpha monomer interconversion rates compared to experimental values. Data is modeled with respect to the apo-Orf9b homodimer 5uM condition. Black line represents reported model fit and values used.

We have also incorporated the following revised text.

“The model parameters k_a_ and k_B_ describe the rate of interchange between the β-sheet and α-helix monomer conformations. These parameters must be estimated by modeling because our assays do not allow us to directly measure the folding rates between these conformations. To identify these values, we performed a scan of k_a_ and k_B_ values that yielded the best agreement between the model and the experimental conditions (Figure 4 - figure supplemental 4).”

The authors build a model that incorporates an α-helix-β-sheet conformational change, but the rate constant for the conversion to the α-helix conformation is required to be second order. Although the authors provide some rationale, I do not find this satisfactorily convincing given the large number of adjustable parameters in the model and the use of manual model fitting. The authors should discuss whether there is any precedence for second-order rate constants for conformational changes in the literature. On page 14, the authors state this rate constant "had to be non-linear in the monomer β-sheet concentration" - how many other models did the authors explore? For example, would αT↔α↔αα↔ββ (i.e., conformational change before dimer dissociation) or α↔βαT↔ββ (i.e., Tom70 binding driving dimer dissociation) be other plausible models for the conformational change that do not require assumptions of second-order rate constants for the conformational change?

We thank the reviewer for their feedback. During our studies, we tested several models prior to the final one presented in Figure 4A. The first model that we tested as described in Figure 4 - Supplemental 3 described ββ↔α↔αT with no conformational change. We tested several models that integrated the existing structural data for both Orf9b and Tom70 and found that while these models could fit individual time series, they did not explain the concentration dependent changes in subsequent time series nor did they explain changes induced by lipid-binding and mutations in VOC.

With respect to the possibilities of αT↔α↔αα↔ββ and α↔βαT↔ββ models, we have revised our manuscript to mention that we did test additional models before we settled on the model that we presented.

“We tested different reaction schemes that incorporated the interconversion between β-sheet to α-helix conformations by considering models that described a conformational change in the homodimer leading to Tom70 binding rather than monomers. None of these models adequately described our experimental results, therefore we continued developing our model as outlined in Figure 4D”

With respect to the second-order rate describing the fold change from β to α, we have added the revised text to the manuscript:

“We initially tested the impact of keeping the rate constant k_a_ first order, just like k_B_ which did yield the sigmoidal behavior we observed in the 5uM apo-homodimer condition. However, this assumption failed to describe the data at other concentrations resulting in a substantial overestimation compared to our experimental results when holding k_B_ at a constant value throughout. We found that when the β-sheet to α-helix rate (k_a_) was made a second order rate constant, we were able to hold the rate constant across all concentrations tested suggesting a non-linearity in the monomer β-sheet concentration.”

While this was surprising to us, we reasoned that a biological explanation for why the conversion from β to α was second order was that the β-monomers may transiently self-associate to cooperatively fold into the α-helical conformation. We did acknowledge this choice to make the β to α parameter non-linear (unlike the α to β conversion which was single order).

We concede that we could not find specific examples describing non-linear kinetics comparable to the system we described in literature, however, such systems have been reported for proteins that exhibit high structural plasticity where transient interactions with another copy of the protein or another protein altogether drive folding changes and we have revised this manuscript to include some additional citations to papers that describe such systems (Zuber et al. 2022; Tuinstra et al. 2008).

Overall, this study progresses the analysis of coupled equilibria and provides insights into Orf9b function.
**Reviewer #1 (Recommendations for the authors):**
(1) What was the unlabeled Orf9b peptide is added to the pre-equilibrated Orf9b-FITC:Tom70 solution as a competitor? Figure 1D illustrates that the competitor was full-length Orf9b.

We have revised the figure to illustrate that in this experiment, the competitor is the unlabeled FITC peptide and not the full length Orf9b sequence

(2) Figure 2B, what is the higher Mw peak from refolded Orf9b homodimer.

We have added the following revised text (highlighted in red) to the manuscript to clarify Figure 2B.

“The SEC elution profile and retention volume of refolded Orf9b directly overlapped with natively folded homodimeric Orf9b and suggested a high recovery of the refolded homodimer with the early eluting peaks corresponding to either a chaperone-bound species (natively folded) or misfolded protein (refolded) as judged by SDS-PAGE (Figure 2B). Together, the overlap in elution peaks corresponding to the folded homodimer suggested a high recovery of the homodimer from the refolding conditions.”

(3) Figure 2C, in the main text, the authors state that "...observed that the refolded homodimer structure closely aligned with the lipid-bound reference structure, which shows that the homodimer fold can be recovered after denaturing". Please provide structural comparison details here, software used? Rmsd and Dali Z-score.

We have added the following revised text (highlighted in red) to the manuscript to clarify Figure 2C.

“Aligning the structure of the Orf9b homodimer (PDB 6Z4U) with our structure of the refolded Orf9b homodimer (9N55) in Pymol resulted in an RMSD of 1.1Å. Further, we also searched our structures of the refolded Orf9b homodimer on the Dali server against the existing structures of the lipid-bound Orf9b homodimer which yielded a Z-score of 2.2 which shows good correspondence between the structures.”

(4) To prove the refolded Orf9b homodimer did not contain lipid, could the authors provide mass spectra data for the refolded Orf9b sample and compare it with the results in Figure 2 - Supplemental 1.

We do not have complete mass spectra data for the refolded homodimer samples, however, we feel that the native mass spectrometry data provides a good orthogonal comparison between natively folded and refolded samples for the presence or absence of lipids. We concede that we only used mass spectrometry to characterize the four peaks that were unique to the natively folded deconvoluted spectra which confirmed that shift in mass relative to the expected homodimer molecular weight corresponded to the two lipids we presented. However, we would expect that performing mass spectrometry on the refolded sample would only further confirm our observations from the crystal structures and the native mass spectrometry.

(5) Have the authors tried to use analytical ultracentrifugation to analyze the Orf9b dimer-monomer equilibrium, given that AUC provides a much more accurate measurement of molecular mass?

We thank the reviewer for this suggestion and agree that AUC could be an additional useful strategy for monitoring the dimer-monomer equilibrium and provide additional validation of the molecule weights of both the monomer and homodimer.

While we have not performed AUC, we have revised our manuscript to include more discussion about the determination of molecular weights by SEC.

“For the Orf9b homodimer, the retention volume was consistent with molecular weight standards based on the expected molecular weight of the homodimer (~21kDa) and the standard (~29kDa). In the case of the Orf9b monomer, although we would expect the retention volume of the monomer (~10.6kDA) to be between the molecular weight standards of 13.4kDa and 6.5kDa, the greater retention volume could be explained by non-specific hydrophobic interactions between the monomeric Orf9b and the column.”

(6) The authors used truncation of 7 C-terminal amino acids to generate an obligate Orf9b monomer for their assays. It would be interesting to mutate residues at the homodimer interface to generate Orf9b monomers rather than deleting residues. For example, mutate 91-96aa (FVVVTV) to negatively charged residues, which will not only disrupt the dimerization interface, but also impair lipid binding. The dimer interface mutant should then be tested in their SPR, FP assays, as well as IFN inhibition assays.

We thank the reviewer for their suggestion and agree that mutation of the 7 C-terminal amino acids into negatively charged residues could be an interesting alternative strategy to generating an obligate Orf9b monomer without the need for truncating the residues. Our choice of using the truncated construct we proposed was driven by our analysis of the structure of the homodimer which reveals that a significant portion of the dimer interface is composed of backbone-backbone hydrogen bonding between the two chains of Orf9b. We reasoned that truncating these residues would be the most effective way to compromise the interface between the two chains and drive a predominantly monomeric behavior, however, compromising the interface with multiple mutations is an intriguing alternative.

**Reviewer #2 (Recommendations for the authors):**

(1) The authors could comment on the slow monomer-dimer exchange observed by SEC and how it fits with their other analysis.

We thank the reviewer for their comment and concede that the slow exchange may be a limitation of this experimental setup. Our observations from our SPR experiments and modeling showed us that the homodimer may be fast to dissociate into monomer given the off rate which would suggest a half-life for the homodimer to be on the order of seconds, however, we still observe a noticeable dimer species on the chromatograms. We initially allowed the diluted samples to reach equilibrium prior to injection onto the analytical sizing column, however, it is possible that the system is still in a pre-equilibrium prior to injection onto the column. This could be driven by interactions between the protein and the column that prevents full dissociation of the homodimer. While this is a limitation, we note that we did not use the Kd value that we determined by non-linear regression fitting to the equilibrium observed on the chromatograms for downstream experiments but instead used the value to get a ballpark estimate for the homodimer Kd which is on the same order as the Kd determined by SPR.

(2) It might be useful to include the rate constants on the reaction arrows of the schematic representation of the models.

We have revised Figure 4D to include the rates for both Orf9b monomer binding to Tom70 and Orf9b binding to Orf9b as derived from the SPR experiments as well as the modeled values for the interconversion between α and β monomers. We also revised Figure 7 to include these values as well as the modeled dissociation rate for homodimer when lipid-bound.

(3) I couldn't find how the sensitivity analysis was performed for the more complex models. Was this the same +/- 10% as per the peptide model?

We used the same +/- 10% sensitivity analysis for the peptide model in the more complex equilibrium model and have revised our manuscript to clearly reflect that.

(4) Further clarification of "inspection of residuals suggested that the fits were accurate". In Figure 1B, the residues look to have systematic errors, perhaps indicating other processes occurring.

We agree that in the SPR kinetic fitting results for the Orf9b peptide binding to Tom70 in Figure 1B that there are some regions where the fit over or under estimates the experimental results. This is partially the result of limitations in the number of different binding models that we can fit in the analysis software which is why we reported using a 1:1 langmuir binding model. It is certainly possible that there may be some additional binding reactions that occur, however, we limited our use of these specific kinetic results to the peptide model that we proposed in Figure 1D. We did note in the manuscript text that it was necessary for us to change the model parameter values to some extent in order to fit our experimental results which may be partially explained by the SPR fitting errors.

“With the parameter set obtained from the 100µM condition, we then held all parameters fixed and simply changed the peptide concentrations in the model to fit the remaining conditions by hand. We note that this process saw the model parameter values change between 3% at the lowest end up to 70% at the highest end from the experimentally derived values but remained within an order of magnitude of the experimental SPR values. We speculate that this arises due to the differences in experimental setup between SPR and FP-based methods of measuring kinetics.”

(5) The manuscript builds logically, but given the sophisticated nature of the system and the modelling could benefit from more clarity/streamlining in the descriptions/illustrations.

We have revised our manuscript in response to both reviewers comments and hope that the clarity of the work is improved as a result.

(6) Figure 4 Supplement 3 - where did the rate constants for Model 1 come from? Was there any attempt to alter them to fit the data better?

We have clarified in the figure description that the rate constants used in Model 1 were the same values used in Figure 4B (but without the interconversion between beta and alpha rates).

“Comparison of kinetic model 1 and 2 in describing experimental results from the kinetic binding assay. Experimental results using 10uM of refolded Orf9b homodimer are shown as rings with the predicted behavior of model 1 (equilibrium exchange) shown as a dark blue line. The predicted behavior of model 2 (equilibrium exchange with a conformational change between β-sheet and ɑ-helical monomers) is shown as the light blue line. Model parameter values were the same as described in Figure 4D and kept constant in both model comparisons.”

(7) What are and [PT] in the second set of equations (page 13)?

[‘PT] refers to the concentration of “fluorescent probe” (Orf9b-FITC) and Tom70.

(8) "Additionally, the fused homodimer association rate (which can be viewed as a rate of tertiary complex formation)" - can the authors provide a mathematical proof for this?

In the case of the fused homodimer kinetic data, we did not develop a separate model to explicitly take into account the differences between using a fused construct versus the WT construct that can dissociate into monomers. We have clarified our interpretation of this in the manuscript.

“Although our model explicitly describes homodimer dissociation into monomers as a requisite step for Orf9b binding to Tom70, we adapted it for the fusion experimental data. In this case, all model parameters other than the association and dissociation kinetics of the fluorescent probe and Tom70 were adjusted to achieve the best agreement with the experimental data. When applied to the fusion homodimer, the parameters describing homodimer dissociation into separate monomers could instead describe the dissociation of the two β-sheet domains away from each other in the tertiary structure but remaining physically linked through the linker region.”

(9) "For Lambda and Omicron, the P10S mutation results in the serine being positioned to form several hydrogen bonds between R13 and the backbone carbonyl of A11 and L48 within the same chain..." is this taken from AlphaFold predicted structures of the mutants? If so, it should be made clear that this is derived from predicted structures. And even so, AlphaFold can be poor at determining structures of mutants, and so there is greater uncertainty in the prediction of the bonds.

For Lambda, Omicron, and Delta mutations, we used Pymol to examine how the placement of mutations could structurally explain the kinetic differences we observed in our model. We have gone back and clarified in the figure description that these predictions are not derived from AlphaFold.

(10) "biological replicates" - is this different protein purifications?

Yes, in this case biological replicates refer to different protein purifications for all variants described and tested.

(11) Are any of the authors involved in the Berkeley Madonna commercial software used in the manuscript? If so, should this be in the conflict of interest statement?

Yes, Michael Grabe is an owner of Berkeley Madonna, and we have updated our conflicts of interest statement to reflect this.